# Likelihood-based Finetuning of Protein Language Models for Few-shot Fitness Prediction and Design

## Abstract

Protein language models (PLMs) implicitly learn distributional constraints on protein sequences upheld over the course of evolution. As a consequence, the sequence and mutation-level likelihoods of such models form effective zero-shot predictors of mutations. Although various schemes have been proposed for exploiting the distributional knowledge captured by PLMs to enhance supervised fitness prediction and sequence design tasks, a lack of head-to-head comparison across different prediction strategies and different classes of PLM has made it challenging to identify the best-performing methods. Our contribution is to extend previously proposed ranking-based loss functions to develop likelihood scoring functions for *family-based* and *masked* PLMs. We demonstrate that in the low-data setting the best configurations outperform the current SOTA approach, which is based on frozen embeddings. Furthermore, we propose ensembling strategies that exploit the strong dependence of the mutational distributions learned by PLMs on sequence context, showing that they can be used to guide efficient optimisation strategies over fitness landscapes.

## 1. Introduction

Natural protein sequences are the result of evolution via natural selection. Protein language models (PLMs) fit to the distribution of natural sequences and therefore learn to implicitly model functional and structural constraints relevant to protein fitness (Gordon et al., 2024). PLM likelihoods form effective zero-shot predictors of the fitness effects of amino acid mutations (Meier et al., 2021; Notin et al., 2022). These distribution learning capabilities are also highly informative for protein sequence design. Mutations assigned high likelihoods by the PLM can be iteratively incorporated

to improve fitness (Hie et al., 2023), or entire sequences can be sampled from generative PLMs (Madani et al., 2023).

In multi-round design scenarios, experimental techniques are often used to generate labelled datasets associating a collection of sequences with measurements of biological properties of interest. Although in some cases these properties are amenable to high-throughput experimental screening methodologies whose resulting datasets are typically large (Rocklin et al., 2017; Tsuboyama et al., 2023), in general, experimental constraints mean that it might only be feasible to generate measurements for tens or hundreds of proteins in each round (Biswas et al., 2021). It is therefore of considerable interest to ask how to best leverage the zero-shot prediction capacities of PLMs alongside small labelled datasets to improve fitness prediction and sequence design.

One popular paradigm for exploiting the information in pre-trained PLMs involves extracting sequence representations and utilising these as (frozen) inputs into task-specific downstream predictive models (or "heads") (Alley et al., 2019; Biswas et al., 2021; Rao et al., 2019; Dallago et al., 2021; Khan et al., 2023; Notin et al., 2023b). However, recent trends in natural language processing have shown the benefits of directly adapting the distributions of models using task-specific labelled data or pairwise preferences (Ouyang et al., 2022; Rafailov et al., 2023) to fine-tune all parameters, thereby fully exploiting the distributional knowledge contained in the original pretrained model.

While pairwise ranking losses have previously been considered to adapt the likelihoods of unconditional autoregressive PLMs (in the context of protein fitness prediction) (Krause et al., 2021; Lee et al., 2023), it remains under-explored whether similar strategies can effectively be applied across other popular classes of PLMs. Here, we introduce likelihood scoring functions for *masked* PLMs, e.g. ESM-1v (Meier et al., 2021) and ESM-2 (Lin et al., 2022), along with recent *family-based* autoregressive models e.g. PoET (Truong Jr & Bepler, 2023) that condition predictions on multiple-sequence alignments (MSA) and often outperform unconditional autoregressive models for fitness prediction (Notin et al., 2023a). For the first time we are able to directly compare fine-tuning strategies for different classes of PLMs, including the current state-of-the-art

[1]Anonymous Institution, Anonymous City, Anonymous Region, Anonymous Country. Correspondence to:

Preliminary work. Under review by the International Conference on Machine Learning (ICML). Do not distribute.

non-parametric architectures for operating on frozen PLM embeddings, e.g. ProteinNPT (Notin et al., 2023b).

In this paper we (i) extend pairwise ranking loss functions to fine-tune the likelihoods of leading zero-shot fitness predictors trained with both masked- and family-based autoregressive language model objectives, (ii) provide direct comparison with SOTA approaches based on frozen embeddings, as well as regression-based fine-tuning approaches, providing compelling empirical evidence that practitioners should make use of ranking-based fine-tuning schemes, regardless of the specific PLM at hand, especially in low-data regimes, and (iii) develop ensemble strategies compatible with these fine-tuning schemes, demonstrating their effectiveness in multi-round Bayesian optimisation settings.

In this paper, we study the modelling problem *in silico* which mimics ground truth values being available from wet lab experiments. We evaluate the performance on a supervised fitness prediction task and a pool-based optimisation task proposed in Notin et al. (2023b). Crucially, unlike their work, we evaluate performance on a diverse subset of ProteinGym landscapes (Notin et al., 2023a) containing multiple mutations, a more realistic and challenging setting.

## 2. Related work

### 2.1. Zero-shot protein fitness prediction

The most successful models for zero-shot prediction of protein fitness effects attempt to predict the likelihood of particular sets of mutations occurring within a natural protein given its evolutionary context. Traditional methods within this category involve statistical models trained directly on multiple sequence alignments (MSAs) for each protein of interest, such as profile models (Hopf et al., 2017), Potts models (Figliuzzi et al., 2016; Hopf et al., 2017) and VAEs (Frazer et al., 2021; Riesselman et al., 2018). More recent generalisations of such methods involve pretraining large PLMs across all natural proteins. For example, ESM-1v (Meier et al., 2021) and ESM-2 (Lin et al., 2022) are trained using a masked language modelling objective, allowing point mutations to be scored by the ratio of probabilities of mutant and wild-type amino acids. Alternatively, autoregressive models can directly compute the likelihood of entire protein sequences, making them more appropriate for scoring sequences containing multiple mutations (Notin et al., 2022; Nijkamp et al., 2023; Madani et al., 2023). However, unconditional PLMs suffer from a lack of context, often requiring fine-tuning to specialise their distributions towards a particular protein family of interest (Madani et al., 2023). As a result, the leading autoregressive models exploit the information in MSAs to improve predictions, either by biasing language model likelihoods with statistics from the MSA, in the case of Tranception (Notin et al., 2022), or by

explicitly conditioning on the MSA (Hawkins-Hooker et al., 2021; Ram & Bepler, 2022; Truong Jr & Bepler, 2023).

### 2.2. Supervised protein fitness prediction

Fitness prediction has been studied as a supervised learning task in many prior works (Rao et al., 2019; Hsu et al., 2022a; Krause et al., 2021). Several works have sought to exploit the pretrained representations of PLMs to improve performance, using either fine-tuned (Rao et al., 2019) or frozen embeddings (Dallago et al., 2021; Notin et al., 2023b). Nonetheless, approaches based on embeddings risk discarding useful distributional information captured in the models' output layers (Krause et al., 2021). The importance of fully leveraging distribution information for fitness prediction is highlighted by the success of 'augmented density' predictors (Hsu et al., 2022a), which combine zero-shot fitness predictions with either one-hot encoded (Hsu et al., 2022a), or embedded (Notin et al., 2023b) representations of input sequences. The state-of-the-art supervised fitness prediction method ProteinNPT (Notin et al., 2023b) combines these strategies, training a bespoke non-parametric Transformer (Kossen et al., 2021) to reason over both zero-shot predictions and associated sequence embeddings to produce fitness predictions.

Methods seeking to adapt the distributions learned by PLMs directly have been less well studied. Rives et al. (2021) propose to use the log-likelihood ratio between mutant and wild-type amino acids as a regression function, fine-tuning the full model. Krause et al. (2021) suggest using a ranking-based loss function to fine-tune autoregressive PLMs, showing improvements over augmented density baselines on a small set of fitness landscapes. A similar ranking-based loss function was proposed for training non-pretrained CNN architectures on fitness datasets in Brookes et al. (2023). Most recently, Lee et al. (2023) apply ranking-based loss functions derived from the literature on large language model alignment (Rafailov et al., 2023) to fine-tune unconditional autoregressive PLMs. The application of ranking-based loss functions to masked PLMs is also considered in concurrent work (Zhao et al., 2024).

### 2.3. Model-guided protein design

Several works have proposed variants of Bayesian optimization (BO) for designing biological sequences, including proteins (Gruver et al., 2021; Jain et al., 2022; Khan et al., 2023; Stanton et al., 2022; Hie & Yang, 2022). It is common to evaluate BO approaches in an unconstrained setting, where sequences are proposed by an optimiser and evaluated with a black-box oracle. However, recent work suggests that designing such biological oracles is a challenging task in itself (Buttenschoen et al., 2024; Surana et al., 2024). An alternative *in silico* evaluation strategy avoids

the challenge of defining a meaningful oracle function by adopting a pool-based optimisation problem formulation over experimentally determined fitness landscapes (Notin et al., 2023b). Another line of work has sought to provide direct experimental validation of approaches combining uncertainty estimates with PLMs, in settings ranging from zero- (Hie et al., 2023) to few-shot design (Biswas et al., 2021), to single-round design with large training sets of sequence-fitness pairs (Li et al., 2023).

Gruver et al. (2021) study various choices of surrogate model for protein design with BO, finding CNN ensembles to be particularly robust to the kinds of distribution shift encountered during online design. More recently, Greenman et al. (2025) studied a range of uncertainty quantification strategies applied to models trained directly on sequences, and on frozen language model embeddings.

## 3. Background

### 3.1. Ranking-based loss functions

While mean squared error has been widely used as a loss function in training sequence-based predictive models of fitness landscapes, two recent works have advocated the use of ranking-based loss functions (Krause et al., 2021; Brookes et al., 2023). In particular, they suggest parameterising a Bradley-Terry model (Bradley & Terry, 1952) with a learned function of the sequence. The Bradley-Terry model represents the probability that a given sequence $x_i$ has higher fitness $y(x_i)$ than another sequence $x_j$ by parameterising a binary classifier via the difference in scores of each sequence under a learned scoring function $s_\theta(x)$:

$$p(y(x_i) > y(x_j)) = \sigma(s_\theta(x_i) - s_\theta(x_j)), \quad (1)$$

where $\sigma$ is the logistic sigmoid function. The model is trained by maximising the likelihood of the complete set of pairwise comparisons between the fitness values of sequences with respect to the parameters $\theta$ of the scoring function. Concretely, given a batch of $B$ sequences $x_1, ..., x_B$, the loss is given by

$$\mathcal{L} = \sum_{i=1}^{B} \sum_{j=1}^{B} -\mathbb{I}(y(x_i) > y(x_j)) \log \sigma(s_\theta(x_i) - s_\theta(x_j)), \quad (2)$$

where $\mathbb{I}$ is an indicator function. In this way, fitness prediction for a dataset of size $N$ is converted from a regression problem with $N$ labels into a binary classification problem with $N \times N$ labels.

### 3.2. Fine-tuning autoregressive PLMs

To use the ranking-based loss functions to fine-tune an autoregressive protein language model, Krause et al. (2021) propose an unconditional sequence log-likelihood score

function:

$$s_\theta(x) = \sum_{i=1}^{L} \log p(x_i | x_{<i}). \quad (3)$$

Since the log-likelihoods of autoregressive protein language models are strong zero-shot predictors of the fitness effects of mutations (Notin et al., 2022), the difference in log-likelihoods used to parameterise the Bradley-Terry model of Equation 1 can produce an effective pairwise classifier at initialisation, which makes this fine-tuning method particularly relevant in low-data regimes.

## 4. Likelihood-based fine-tuning of masked and family-based PLMs

We describe below how pairwise ranking losses can be extended to fine-tune the likelihoods of other widely used classes of protein language models, masked language models and family-based autoregressive models. These extensions are derived from the principle that fine-tuning strategies should exploit as far as possible the properties of models that lead to strong zero-shot performance (Krause et al., 2021). We enforce this principle by suggesting an appropriate choice of the scoring function $s_\theta$ used to parameterise the Bradley-Terry model in each case.

### 4.1. Conditional scoring functions

Previous applications of the ranking-based loss to fine-tune PLMs have focussed on unconditional autoregressive models. However, these models often underperform other classes of model including conditional autoregressive models and masked language models in fitness prediction settings (Notin et al., 2023a). We therefore propose new likelihood scoring strategies, amenable to the Bradley-Terry model, to accommodate these newer PLM classes. To do so, we incorporate the additional conditioning information $c$ exploited by these models into the conditional scoring function $s_\theta(x, c)$:

$$p\big((y(x_i) > y(x_j))|c\big) = \sigma(s_\theta(x_i, c) - s_\theta(x_j, c)). \quad (4)$$

Below, we will consider cases where $c$ represents either a wild-type sequence or a multiple sequence alignment (MSA), since conditioning on evolutionary context is especially effective in fitness prediction (Truong Jr & Bepler, 2023). Note the same approach could be applied to models which condition on protein structure (Hsu et al., 2022b).

### 4.2. Scoring functions for masked PLMs

Masked language models do not define a sequence-level likelihood, meaning that it is not immediately obvious how to define a scoring function for the Bradley-Terry model. Meier et al. (2021) proposed a variety of strategies for zero-shot scoring of mutants using the likelihoods assigned to

sets of mutations by masked language models. We propose to use these zero-shot scoring-functions to parameterise the Bradley-Terry model in Equation 4, allowing the models to be fine-tuned with ranking-based losses, similar to other concurrent work (Zhao et al., 2024). Concretely, for single mutation ProteinGym landscapes, we utilize the 'wild-type marginals' (wt-marginals) scoring function. Under this strategy the score for a mutated sequence is given by the summation of the log-likelihood ratios between mutated and wild-type amino acids across mutated positions, given the unmasked wild-type sequence as input:

$$s_\theta(x, x^{\text{wt}}) = \sum_{i:x_i^{\text{wt}} \neq x_i} \log p(x_i|x^{\text{wt}}) - \log p(x_i^{\text{wt}}|x^{\text{wt}}) . \quad (5)$$

We choose the marginal strategy for its combination of computational efficiency and strong performance as a zero-shot scoring function (Meier et al., 2021). Since all sequences are scored under the residue distributions obtained by inputting the wild-type sequence to the model, a set of mutated sequences of arbitrary size can be scored using a single forward pass, making it extremely efficient.

For landscapes with multiple mutations, the wt-marginal strategy assumes an additive likelihood function over mutations, and may not adequately capture the epistasis effects, i.e. where the combined effect of mutations at different positions is not simply the additive result of their individual effects. For this reason, we explore additional marginal strategies in Section 5.2 Result 4 (described further in Appendix B.4).

### 4.3. Scoring functions for family-based PLMs

Family-based PLMs represent the conditional distribution over family members given a subset of other family members (Rao et al., 2021; Hawkins-Hooker et al., 2021; Ram & Bepler, 2022; Truong Jr & Bepler, 2023). These models have proved effective as zero-shot fitness predictors, due to their ability to explicitly condition on evolutionary context to predict the effects of mutations.

In this paper we work with PoET (Truong Jr & Bepler, 2023), which models entire protein families autoregressively. To produce predictions given a mutant sequence $x$ and an MSA $M = \{m^{(1)}, ..., m^{(N)}\}$ of homologues of a wild-type sequence $x^{\text{wt}}$, PoET computes the likelihood of the mutant $x$ conditional on the MSA $M$. To exploit this capacity to condition on family members during fine-tuning, we condition the autoregressive scoring function in Equation 3 on the sequences in the MSA:

$$s_\theta(x, M) = \sum_{i=1}^{L} \log p(x_i|x_{<i}, M) . \quad (6)$$

Since PoET operates natively on unaligned sequences and is sensitive to alignment depth, we subsample a small set of

sequences from the MSA and discard gaps before feeding them into the model, following (Truong Jr & Bepler, 2023) (details in Appendix A.2).

To increase the efficiency of fine-tuning PoET, in practice we cache a single set of hidden layer representations obtained by passing the subsampled MSA $M$ through the model, and fine-tune only the mapping between these frozen representations and the sequence likelihoods (Appendix A.3). This effectively decouples the encoding of prior context from the decoding of future amino acids given this context.

### 4.4. Uncertainty quantification with evolutionary context ensembles

The amino acid output distributions learned by PLMs depend heavily on sequence context, which language models must use to infer the (structural or functional) constraints determining native amino acid identities. We propose to exploit this property to build ensembles of fine-tuned PLMs, in which each ensemble member sees a different, but functionally equivalent, sequence context. Concretely, we do this for both family-based and masked PLMs: For *family-based* models we fine-tune an ensemble of PoET models, for each fitness landscape we sub-sample a set of $K$ input MSAs $M_{1:K}$ from the full MSA associated with the wild-type sequence. We then fine-tune a separate set of parameters to minimise the loss conditioned on each MSA, producing $K$ sets of parameters, each specialised to a single input MSA (further details are provided in Appendix A.2). To score sequences, we define the ensemble scoring function as:

$$s_{\theta_{1:K}}(x, \{M_{1:K}\}) = \frac{1}{K} \sum_{k=1}^{K} s_{\theta_k}(x, M_k). \quad (7)$$

This procedure extends the practice of MSA ensembling used to improve the zero-shot predictions of MSA-based PLMs (Truong Jr & Bepler, 2023), to the supervised setting.

For *masked* models, to achieve a similar effect with no MSA representation, we instead sample a set of $K$ input masks, and fine-tune a separate set of parameters for each input mask, exploiting the intuition that differently masked sequences remain functionally equivalent, but may nonetheless produce different outputs when passed through the model (Appendix A.4). We demonstrate this approach in Section 6.2 Result 1 for ESM-2 (Lin et al., 2022).

### 4.5. Relationship to preference learning for LLMs

Direct preference optimisation (DPO) (Rafailov et al., 2023) is a recently proposed method for aligning large language models (LLMs) using datasets of human preference data. DPO also uses scoring functions from pretrained models to parameterise a Bradley-Terry model. Instead of parameterising a classifier directly via differences in log likelihoods,

DPO uses the difference in scaled log likelihood *ratios* between the fine-tuned model and a frozen reference model. Thus the probability that a completion $x_1$ is preferred to a completion $x_2$ given an input prompt $c$ is modelled as:

$$p_\theta(x_1 \succ x_2|c) = \sigma\left(\beta\log\frac{p_\theta(x_1|c)}{p_{\text{ref}}(x_1|c)} - \beta\log\frac{p_\theta(x_2|c)}{p_{\text{ref}}(x_2|c)}\right). \tag{8}$$

In our notation, the DPO preference model therefore amounts to a particular choice of scoring function $s_\theta(x, c) = \beta\log\frac{p(x|c)}{p_{\text{ref}}(x|c)}$. Assuming an autoregressive decomposition of $p(x|c)$, this scoring function is equivalent to the conditional autoregressive scoring function in Equation 6 if the reference model is constant and $\beta = 1$.

The non-constant reference model in DPO imposes a KL penalty on the deviation between the fine-tuned $p_\theta$ and the reference model, which helps prevent collapse in the fine-tuned distribution (Rafailov et al., 2023). Although some recent work has reported success in adapting DPO to the protein fitness prediction setting (Lee et al., 2023), in our own experiments we did not find this regularisation necessary to achieve good performance. We hypothesise that this is because we do not require generations from the model, unlike typical applications of DPO.

## 5. Experiment: Low-n fitness prediction

### 5.1. Experiment details

**Protein fitness landscapes**   We study the performance of fitness prediction strategies on mutational landscapes from ProteinGym (Notin et al., 2023a). Each landscape contains a set of protein sequences together with experimentally determined fitness values. The protein sequences within a landscape contain a small number of mutations relative to the 'wild-type' protein, and the fitness values are quantitative measurements of a functional property associated with the wild-type. We utilise two subsets of ProteinGym: the first is the validation set of 8 representative single-mutant landscapes selected by Notin et al. (2023b). The second is a set of 5 landscapes containing multiple mutations, that constitutes a non-redundant set of diverse landscapes available in ProteinGym (Appendix A.1).

In contrast to prior work (Notin et al., 2023b), we focus explicitly on the low-data setting. For each landscape, we train all methods on $n = 32, 128$ or $512$ sequences randomly sampled from the landscape and evaluate on either 2000 (for single-mutant landscapes) or 5000 (for multiple-mutant landscapes) randomly sampled held-out sequences. An additional set of 128 randomly sampled sequences is used as a validation set to perform early stopping. For each landscape, and each $n$, we generate three sets of random splits, and report test set Spearman correlation averaged across

the three splits. For models trained with ranking losses, the Spearman correlation is computed between the scoring function $s_\theta(x, c)$ and the ground truth fitness values.

Prior work has also considered non-random splits (Dallago et al., 2021; Notin et al., 2023b), including splits that test generalisation to mutations at unseen positions. We note that such generalisation is required in the low-data setting, as not all positions will be represented in the small training sets. We therefore present our main results on randomly generated splits, similar to (Hsu et al., 2022a; Krause et al., 2021). However, in Section 5.2 Result 3 we also assess generalisation by computing metrics on subsets of the test sets containing mutations at positions for which no mutations were present in the training set sequences.

**Fitness prediction strategies**   We evaluate the performance of the fine-tuning strategies introduced in Section 4 on the selected landscapes. To attain an understanding of the effectiveness of these strategies across different classes of PLM, we apply them to the masked language model ESM-1v (Meier et al., 2021) and ESM-2 (Lin et al., 2022), and the family-based autoregressive model PoET (Truong Jr & Bepler, 2023). In each case, the model is fine-tuned by parameterising the Bradley-Terry model of Equation 1 via the corresponding scoring functions in Section 4, and minimising the ranking loss in Equation 2. As an ablation, we compare to a mean square error (MSE) loss applied to the same scoring function.

Additionally, we compare to two further sets of baselines representative of widely used approaches, that either (i) fine-tune PLMs by adding a regression head (Rao et al., 2019), or (ii) train new models on top of frozen language model embeddings (Notin et al., 2023b). In the first case, we add a linear regression head to ESM-1v, ESM-2 and PoET, and fine-tune all parameters with an MSE loss (additional details provided in Appendix A.2.1). Further, in Appendix B.1 we present an ablation using the ranking loss in Equation 2 applied to the regression target. As the leading example of the second class of approaches, we compare against Protein-NPT (Notin et al., 2023b), a state-of-the-art model operating on frozen language model embeddings. As additional baselines, we include the 'augmented density' strategies used as baselines in Notin et al. (2023b). These are regression models, taking as input the zero-shot predictions of a PLM as well as either a one-hot representation of the mutated sequence (Hsu et al., 2022a), or an embedding extracted from a PLM (further details in Appendix A.6). We refer to these distinct choices of augmented density representation as 'OHE augmented' (OHE aug.) and 'Embedding augmented' (Emb. aug.) respectively, following code made available in Notin et al. (2023b). Hyperparameters are selected based on performance on the single mutant set, consistent with the practice used for ProteinNPT and associated baselines.

*Table 1.* Fitness prediction in low data settings, comparing Spearman correlation (higher is better) of masked and family-based PLM models with varying scoring strategies and loss functions. Evaluated on 8 single mutant landscapes and 5 multiple mutant landscapes from ProteinGym. ProteinNPT and baseline models use a frozen base model to produce embeddings (base model provided in parentheses).

| Model Name | Scoring function | Loss | single-mutants | | | multi-mutants | | |
|---|---|---|---|---|---|---|---|---|
| | | | $n = 32$ | $n = 128$ | $n = 512$ | $n = 32$ | $n = 128$ | $n = 512$ |
| ESM-1v (650M) | linear head | mse | 0.263 | 0.415 | 0.535 | 0.494 | 0.637 | **0.771** |
| | wt-marginals | **ranking** | **0.479** | **0.552** | **0.641** | **0.577** | **0.642** | 0.753 |
| ESM-2 (650M) | linear head | mse | 0.280 | 0.398 | 0.535 | 0.427 | 0.596 | 0.743 |
| | wt-marginals | **ranking** | **0.455** | **0.530** | **0.627** | **0.593** | **0.651** | **0.758** |
| PoET | linear head | mse | 0.443 | 0.553 | 0.646 | 0.571 | 0.714 | 0.793 |
| | likelihood | **ranking** | **0.513** | **0.591** | **0.672** | **0.667** | **0.737** | **0.806** |
| ProteinNPT (MSAT) | | mse | 0.415 | 0.533 | 0.637 | 0.517 | 0.692 | 0.791 |
| ProteinNPT (ESM-1v) | | mse | 0.410 | 0.497 | 0.607 | 0.438 | 0.645 | 0.769 |
| Emb. aug. (MSAT) | | mse | 0.424 | 0.507 | 0.553 | 0.581 | 0.696 | 0.764 |
| Emb. aug. (ESM-1v) | | mse | 0.451 | 0.505 | 0.550 | 0.440 | 0.624 | 0.702 |
| OHE aug. (MSAT) | | mse | 0.429 | 0.467 | 0.496 | 0.616 | 0.684 | 0.763 |
| OHE aug. (ESM-1v) | | mse | 0.466 | 0.502 | 0.526 | 0.460 | 0.566 | 0.711 |
| OHE | | mse | 0.144 | 0.314 | 0.488 | 0.268 | 0.473 | 0.664 |

## 5.2. Results

**Result 1: Ranking-based fine-tuning outperforms regression fine-tuning** We first focus on the comparison between ranking-based fine-tuning and regression-based fine-tuning in Table 1 (top), *using the same models*: i) For PoET, ranking-based likelihood fine-tuning performs best across all dataset sizes for single- and multi-mutant landscapes. Regression-based fine-tuning is nonetheless a strong baseline, performing slightly better than the best ProteinNPT configuration. ii) For masked models, ESM-1v and ESM-2, ranking-based likelihood fine-tuning performs much better than regression-based fine-tuning across both the single- and mutli-mutant landscapes in all dataset sizes except the $n = 512$ multi-mutant setting. As discussed in Section 4, in the $n = 512$ multi-mutant setting this result is likely due to the linear regression head accounting for epistasis effects better than the wild-type marginal likelihood scoring rule. Further evidence of this is provided in Section 5.2 Result 4. Finally, as one expects, we observe the gap in performance between the two fine-tuning approaches shrinks with increasing $n$.

For completeness, in Appendix B.1 Table 6 we ablate other possible configurations: i) the MSE loss applied to the PLM likelihood score function, and ii) the ranking loss in Equation (2) applied to the PLM with a linear regression head. Whilst an interesting ablation, the application of MSE loss directly to the likelihoods does not perform well. However, the ranking loss applied to the regression target is a strong baseline, especially in the $n = 128$ and 512 multi-mutant settings, demonstrating improved performance for ESM-1v and ESM-2 over the standard MSE fine-tuning.

**Result 2: Ranking-based fine-tuning outperforms models trained on frozen embeddings** We next focus on the comparison between the best-performing fine-tuning schemes Table 1 (top) and the set of ProteinNPT baselines (bottom) that utilise frozen embeddings from a base model (Notin et al., 2023b). The full results are provided in Appendix Table 7. ProteinNPT (MSAT) is the current SOTA for ProteinGym single-mutant landscapes. We demonstrate here that ranking-based fine-tuning of PoET outperforms ProteinNPT across all settings, with the gap largest in the lowest data regimes, suggesting that directly adapting the likelihoods of the pretrained PoET model is particularly impactful for maximising performance in limited data regimes. Notably this is not simply by virtue of PoET producing better zero-shot predictions: on the single mutant datasets, the zero-shot ESM-1v predictions used by ProteinNPT (ESM-1v) outperform those produced by PoET (0.437 vs 0.417 evaluated on the $n = 128$ data splits). Masked PLMs ESM-1v and ESM-2 fine-tuned via the wt-marginal likelihood ranking strategy also outperform ProteinNPT on the single-mutant datasets, but performs worse on the multi-mutant datasets, likely due to the limited expressivity of the scoring rule, as discussed further in Section 5.2 Result 4.

**Result 3: Ranking-based fine-tuning generalises to unseen positions** In Table 2 we assess the capability of the fine-tuning methods to generalise to mutations at unseen positions in the test set sequences. That is, random splits provide an estimate of performance on heldout data. However, similar mutations can occur in both train and test sets (e.g. related amino acid substitutions at the same position), meaning that measuring performance on predicting the effects of these mutations does not necessarily test a model's

*Table 2.* Single-mutant Spearman correlations for test set mutations at seen vs unseen residue positions ($n$=128). Test set sequences are assigned to the unseen set if they contain mutations at residue positions at which none of the training set sequences have mutations. ProteinNPT (referred to as PNPT) uses a frozen base model provided in parentheses.

| Model | Scoring fn. | Loss | Seen | Unseen |
|---|---|---|---|---|
| ESM-1v (650M) | linear head | mse | 0.460 | 0.331 |
| | wt-marginals | rank | **0.592** | **0.509** |
| ESM-2 (650M) | linear head | mse | 0.453 | 0.297 |
| | wt-marginals | rank | **0.568** | **0.455** |
| PoET | linear head | mse | 0.571 | 0.517 |
| | likelihood | rank | **0.612** | **0.549** |
| PNPT (MSAT) | - | mse | 0.563 | 0.462 |
| PNPT (ESM-1v) | - | mse | 0.529 | 0.420 |

*Table 3.* Masked-marginal scoring strategies for ESM-2 (650M) with ranking loss applied to the five multi-mutant landscapes. "Steps" specifies the number of inference steps required per batch of training sequences; $B$ = batch size; $M$ = number of mutations; and $K$ = masked-modulo constant. Full results in Appendix B.4.

| Scoring function | Steps | $n$=32 | $n$=128 | $n$=512 |
|---|---|---|---|---|
| wt-marginals | 1 | **0.593** | 0.651 | 0.758 |
| mt-marginals | $B$ | 0.398 | 0.591 | 0.750 |
| masked-marginals-a | $B$ | 0.559 | 0.651 | 0.766 |
| masked-marginals-b | $2MB$ | 0.492 | 0.607 | 0.744 |
| masked-marginals-c | $MB$ | 0.493 | 0.598 | 0.751 |
| masked-modulo | $KB$ | 0.534 | **0.661** | **0.774** |

*Table 4.* Ensemble PLM models ($n = 32, 128$) evaluated on five multi-mutant landscapes from ProteinGym. PNPT with dropout uses 25 Monte Carlo simulations. ESM-2 ensemble uses the wt-marginal scoring strategy; PoET MSA-ensemble uses likelihood, and both use a ranking loss function and 5 ensemble members.

| PLM Model | $n = 32$ | $n = 128$ |
|---|---|---|
| PNPT (MSAT) | 0.517 | 0.692 |
| PNPT (MSAT) with dropout | 0.512 | 0.696 |
| ESM-2 | 0.593 | 0.651 |
| ESM-2 ensemble | 0.621 | 0.683 |
| PoET | 0.667 | 0.736 |
| PoET MSA ensemble | **0.696** | **0.757** |

capacity for generalisation (Notin et al., 2023b). This issue is somewhat mitigated in our setup by the choice of relatively small training sets, and the emphasis on multi-mutant datasets, where the degree of overlap between train and test sets is typically lower.

We provide results here for the single mutant landscapes, and a complete table of results for the multi-mutant land-

scape in Appendix B.3 Table 8). We report the performance for mutations in the $n = 128$ test sets occurring at positions at which no mutations were present in the training set sequences. While there is a clear and expected drop in performance at these unseen positions, ranking-based fine-tuning directly on the likelihoods for all models demonstrates the best out-of-distribution behaviour at unseen positions.

**Result 4: Masked scoring strategies to capture epistasis effects** In Table 3 we empirically compare masked PLM scoring strategies on the five multi-mutant ProteinGym landscapes (the full results are provided in Appendix B.4 Table 9). These more expressive strategies proposed in Meier et al. (2021) and Johnson et al. (2024) utilise additional forward inference steps per sequence in order to better capture the effects of multiple mutated residues, known as epistasis, i.e. where the combined effect of mutations at different residues may not simply be the additive result of their individual effects as assumed by the wt-marginal strategy.

For the first time, we can demonstrate that whilst not all additional compute improves over the highly efficient wt-marginal strategy, the "modulo" masking strategy (Johnson et al., 2024) outperforms all others with $n = 128$ or 512, but requires $K.B$ times more forward passes, where $K = 4$ or 8 depending on landscape (specified in Appendix B.4).

## 6. Experiment: Multi-round sequence design

### 6.1. Experiment details

We next ask whether the improvements in fitness prediction translate to benefits in a multi-round sequence design. To do so, we follow the evaluation protocol introduced by Notin et al. (2023b), with minor modifications. Sequence design is formulated as a pool-based optimisation task over the sequences in an empirical fitness landscape. For a given landscape, the goal is to retrieve as many high-scoring sequences as possible over the course of 10 optimisation rounds (recall). In each round, the model's predictions are used to guide the acquisition of a batch of 100 sequences from a pool of candidate sequences. The pool of candidate sequences is either the complete landscape, or, in the case of the multiple mutant landscapes, a randomly selected subset of 5000 sequences. Before the first round, models are fine-tuned on 100 sequences randomly sampled from the landscape. All experiments are run on three random seeds.

We follow Notin et al. (2023b) in using ensembling strategies to derive uncertainty estimates which can be used to guide the selection of candidates from the pool within the framework of Bayesian optimisation (BO). We make use of the upper confidence bound (UCB) acquisition function. We note that our use of a ranking loss means that our ensemble surrogates are preferential surrogates and, as such, alternative (preferential) acquisition strategies from the field

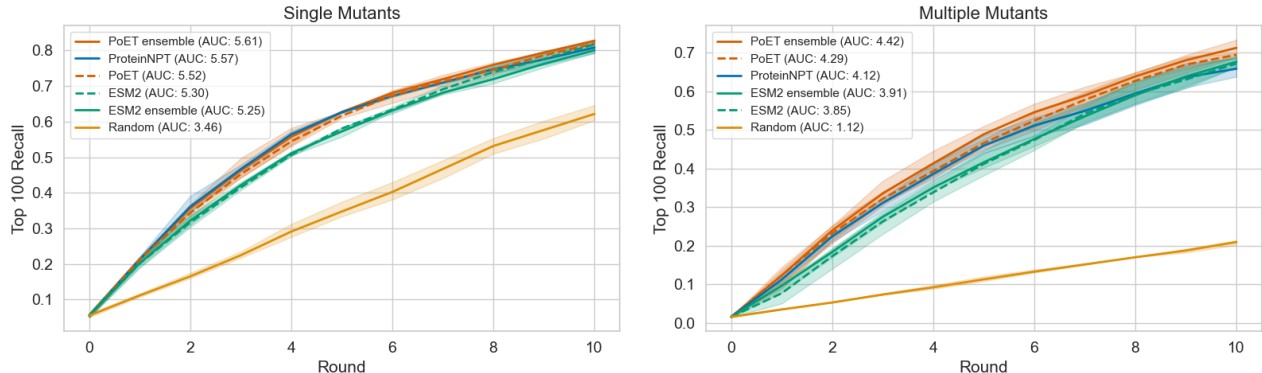

*Figure 1.* Recall results for multi-round pool-based optimisation task. PoET MSA-ensemble and ESM-2 ensemble both likelihood ranking-based fine-tuned, along with their greedy single model counterparts (as dashed lines) for 8 single mutant landscapes (left) and 5 multiple mutant landscapes (right). Additional plots are provided in Appendix Figure 2 (masked-ensembles), Figure 3 (PoET ensembles), and Figure 4 (ProteinNPT baselines). AUC refers to the area under the curve (higher is better).

of preferential BO (González et al., 2017) might better exploit the model's pairwise classification uncertainty. At each round, we rank all remaining sequences in the pool by their acquisition values, and select the top 100 to add to the training set. For ProteinNPT we use Monte Carlo dropout (Gal & Ghahramani, 2016) to produce uncertainty estimates.

### 6.2. Results

**Result 1: Masked and family-based PLM-ensembles** In Table 4 we compare the fitness prediction performance of the ensemble models introduced in Section 4.4 to their single PLM model counterpart, averaged over the five multi-mutant landscapes. Specifically, unlike ProteinNPT (referred to as PNPT in the table), we see that the additional context provided by the evolutionary MSA in the PoET ensemble and the masked-ensemble ESM-2 model improves the low-n fine tuning performance. Full results for each scoring strategy and loss functions are provided in Appendix B.7 Table 12.

**Result 2: Sequence design guided by PLM-ensembles** In Figure 1 we present pool-based optimisation results (recall curves) guided by our ensemble models for PoET (orange) and ESM-2 (green) compared to the ProteinNPT (MSAT) baseline (blue). Results are averaged over 8 single mutation landscapes (left) and 5 multiple mutant landscapes (right). Additional baseline design curves are provided in Appendix B.8.3, and a table of all method's AUC top 30% recall and top 100 recall is provided in Appendix Table 13.

Across both sets of landscapes, the PoET ranking-based MSA-ensemble outperforms all other methods. In general, the recall design curves show similar trends to the supervised results. Ranking-based fine-tuning outperforms regression-based fine-tuning, and leveraging our novel ensemble strategies leads to the best overall design performance. However, the single PLM models with greedy ac-

quisition strategy also provide a strong baseline.

While recall of high-fitness sequences saturates for the single mutant landscapes, it improves steadily for the multiple mutant landscapes, since the starting pools are larger, and it is not possible to reach perfect recall within the fixed budget of acquisitions. Design curves for each individual landscape are provided in Appendix B.8.4 (singles) and Appendix B.8.5 (multiples). The relative ordering of the methods is reasonably stable across individual landscapes, although there are some cases where the non-PLM baselines perform comparably to the best-performing methods, suggesting these landscapes may contain noisy or otherwise difficult-to-predict fitness labels (Notin et al., 2023b).

## 7. Conclusion

The ability of language models to learn distributional constraints governing natural protein sequences makes them powerful zero-shot predictors of the effects of mutations on protein function. We demonstrate that these learned distributions can be rapidly adapted via likelihood-based fine-tuning from as few as 32 experimental measurements - of the order of a typical low batch size in biological experiments. In this paper, we extend existing ranking-based scoring functions to the masked- and autoregressive family-based PLM settings via explicitly conditioning on evolutionary information. We surpass the leading baseline approaches, and as such, provide strong empirical evidence that practitioners should make use of ranking-based losses regardless of PLM choice, especially in low-data regimes. Further, we go beyond existing literature, providing in-depth analysis on out-of-distribution and epistasis effects for multiple mutant landscapes. Finally, we demonstrate ensembling strategies that are compatible with likelihood fine-tuning, demonstrating their effectiveness in multi-round sequence design tasks.

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

## A. Appendix

### A.1. Fitness landscapes from ProteinGym

We use the set of 8 single-mutant landscapes selected for ablations and hyperparameter selection by (Notin et al., 2023b). The names of these landscapes in ProteinGym are:

- BLAT_ECOLX_Jacquier_2013
- CALM1_HUMAN_Weile_2017
- DYR_ECOLI_Thompson_2019
- DLG4_RAT_McLaughlin_2012
- REV_HV1H2_Fernandes_2016
- TAT_HV1BR_Fernandes_2016
- RL40A_YEAST_Roscoe_2013
- P53_HUMAN_Giacomelli_WT_Nutlin

We additionally select a set of 5 of the most diverse multi-mutant landscapes in ProteinGym. To select these landscapes, we identified the landscapes with the largest number of mutations in ProteinGym, and discarded redundant landscapes (for example the GFP landscapes of (Gonzalez Somermeyer et al., 2022) are landscapes of close homologues of the GFP protein whose landscape was reported by (Sarkisyan et al., 2016). We therefore include only the latter.

The selected multi-mutant landscapes are:

- PABP_YEAST_Melamed_2013
- CAPSD_AAV2S_Sinai_2021
- GFP_AEQVI_Sarkisyan_2016
- GRB2_HUMAN_Faure_2021
- HIS7_YEAST_Pokusaeva_2019

Table 5. Additional details of the 8 single mutation, and 8 multiple mutation landscapes we use from ProteinGym. wt-length is the number residues in the wild-type protein. Mutations refers to whether the landscape contains single mutant sequences, or the range of the number of mutations present in the landscape. Fitness score distribution is provided as mean and std.

| Landscape Name | wt length | Mutations | Num of seqs | Fitness Score Distr. |
|---|---|---|---|---|
| BLAT_ECOLX_Jacquier_2013 | 286 | Single | 989 | $-1.558 \pm 1.952$ |
| CALM1_HUMAN_Weile_2017 | 149 | Single | 1813 | $0.742 \pm 0.365$ |
| DLG4_RAT_McLaughlin_2012 | 724 | Single | 1,576 | $-0.172 \pm 0.406$ |
| DYR_ECOLI_Thompson_2019 | 159 | Single | 2,363 | $-0.391 \pm 0.742$ |
| P53_HUMAN_Giacomelli_2018_WT_Nutlin | 393 | Single | 7,467 | $-0.020 \pm 1.035$ |
| REV_HV1H2_Fernandes_2016 | 116 | Single | 2,147 | $-0.121 \pm 0.218$ |
| RL40A_YEAST_Roscoe_2013 | 128 | Single | 1,195 | $-0.265 \pm 0.345$ |
| TAT_HV1BR_Fernandes_2016 | 86 | Single | 1,577 | $-0.116 \pm 0.197$ |
| CAPSD_AAV2S_Sinai_2021 | 735 | Multiple (1-28) | 42,328 | $-1.226 \pm 3.045$ |
| GFP_AEQVI_Sarkisyan_2016 | 238 | Multiple (1-15) | 51,714 | $2.658 \pm 1.059$ |
| GRB2_HUMAN_Faure_2021 | 217 | Multiple (1-2) | 63,366 | $-0.793 \pm 0.467$ |
| HIS7_YEAST_Pokusaeva_2019 | 220 | Multiple (1-28) | 496,137 | $0.619 \pm 0.449$ |
| PABP_YEAST_Melamed_2013 | 577 | Multiple (1-2) | 37,708 | $0.524 \pm 0.391$ |

### A.2. Hyperparameter details

Hyperparameters for the fine-tuning methods were selected based on performance on the single mutant set, consistent with the practice used to select hyperparameters for the baselines from ProteinNPT. We report metrics obtained when using these hyperparameters on both single-mutant and multiple-mutant landscapes for each method.

ESM-1v, ESM-2 and PoET models were fine-tuned using the Adam optimizer (Kingma & Ba, 2015) using gradient accumulation with an effective batch size of 32. Learning rates for regression-based and ranking-based fine-tuning were selected separately in each case after after a sweep over the values $1e-4, 3e-5, 1e-5$ on the 8 single mutant landscapes. For ESM models, we computed the loss by scoring all sequences using the wt-marginal strategy. In the fitness prediction experiments, the models were trained for 50 epochs. During training on each landscape the Spearman correlation, computed on a separate validation set of 128 sequences from the landscape, was used to determine the epoch whose checkpoint should be used to produce predictions on the test set.

### A.2.1. REGRESSION HEADS

Linear regression heads were added to embeddings extracted from ESM-1v, ESM-2 and PoET. In the former case, we averaged embeddings across the sequence dimension before inputting them to the regression head, in the latter case we used final token embeddings.

### A.2.2. ENSEMBLES

Contextual ensemble models of size 5 were used for both ESM-2 and PoET. During design, the ensemble members were trained for 20 epochs each round. In each round all ensemble members were reinitialised from the pretrained model and retrained on the latest training dataset.

### A.3. Decoder-only fine-tuning of PoET

PoET parameterises a sequence of conditional distributions over the amino acids in a set of protein sequences in the same family. The model represents the joint likelihood of a set of sequences $M = \{m^{(1)}, ..., m^{(N)}\}$, via an autoregressive factorisation over sequences and over positions within each sequence:

$$p(M) = \prod_i p(m^{(i)}|m^{(<i)}) = \prod_{ij} p(m_j^{(i)}|m_{<j}^{(i)}, m^{(<i)}) \,. \tag{9}$$

To parameterise this distribution, PoET uses a causally masked Transformer architecture, which maps from previous amino acids to logits for the current amino acid. Conceptually, this function can be decomposed into two stages: first the entire history of previous sequences $m_{<i}$ is encoded into a sequence of embeddings $H_{<i} \in \mathbb{R}^{L_{<i} \times D \times E}$, where $D$ is the number of layers and $E$ is the embedding dimension, via a stack of causally masked layers:

$$H_{<i} = f_\theta(m^{(<i)}) \,. \tag{10}$$

The current sequence $m_i$ is then decoded by a function which maps these prior sequence embeddings and previous amino acids in the current sequence to logits for each position $j$:

$$\text{logit}_{ij} = g_\theta(m_{<j}^{(i)}, H_{<i}) \,. \tag{11}$$

To fine-tune PoET from fitness data, we propose to fine-tune only the weights of the function $g$, representing the 'decoding' of the current sequence given its context. To achieve this, we first clone the PoET weights, producing a set of 'encoder' weights $\phi$ and a set of 'decoder' weights $\theta$. We use the frozen encoder weights to produce an embedding $H \in \mathbb{R}^{L_M \times D \times E}$ of the input MSA sequences: $H = f_\phi(\{m^{(1)}, ..., m^{(N)}\})$, where $L_M$ is the total length of all sequences in the input MSA. We then fine-tune the weights $\theta$ of the cloned 'decoder' to minimise the cross-entropy loss of Equation 2 on the labelled data. Concretely, the scoring function used to parameterise the Bradley-Terry model becomes:

$$s_\theta(x, M) \equiv s_\theta(x, H) = \sum_i \log p_\theta(x_i|x_{<i}, H) \tag{12}$$

To maximise computational efficiency, the MSA embeddings $H$ are pre-computed before the start of the fine-tuning process, and remain frozen throughout.

### A.4. ESM-2 ensembling strategy

To fine-tune an ensemble of models given a single ESM-2 checkpoint, we randomly sampled a set of 5 masks. Within each mask, each sequence position had a $15\%$ probability of being masked. We fine-tuned one model for each mask, by using the correspondingly masked wild-type sequence $\tilde{x}_k^{\text{wt}}$ as input to the model, instead of the unmasked wild-type sequence. The ensembled scoring function used to generate predictions is:

$$s(x, x^{\text{wt}}) = \frac{1}{K} \sum_k s(x, \tilde{x}_k^{\text{wt}}) \tag{13}$$

### A.5. PoET MSA subsampling

For PoET, in both single-model and ensemble configurations, we sampled context sequences from the same filtered MSAs used to extract MSA Transformer embeddings for ProteinNPT. These MSAs are generated from the full MSAs provided with ProteinGym by running hhfilter (Steinegger et al., 2019), requiring a minimum coverage of 75% and a maximum sequence identity of 90%. Subsequently, we use weighted sampling as described in Truong Jr & Bepler (2023) to select sequences to pass as context to PoET, up to a maximum context length of 8192 tokens. The MSA is encoded using a frozen copy of the PoET model into a set of cached hidden representations, as described in Appendix A.3. When ensembling, a separate MSA is sampled for each ensemble member, and held fixed during the fine-tuning of that ensemble member.

### A.6. Baseline models

ProteinNPT, the embeddings augmented (Emb. aug.) baselines, and the one-hot encoding augmented (OHE aug.) baselines were all run using the code released by Notin et al. (2023b). The one-hot and embedding augmented models both use the strategy from Hsu et al. (2022a), combining the zero-shot predictions from a pretrained model with sequence features in a regression framework. They differ in the way sequence features are extracted: in the former case, ridge regression is performed directly on the one-hot encoded sequences. In the latter case, PLM embeddings are used to featurise the sequences. We refer to Notin et al. (2023b) for further details.

For the fitness prediction experiments, separate ProteinNPT models were trained for 2000 and 10000 steps, and the results of the best-performing model were reported. The other baselines appeared to benefit more from longer training and were trained for 10000 steps, as in Notin et al. (2023b). For design experiments, we used the Monte Carlo dropout uncertainty quantification strategy proposed by Notin et al. (2023b) for both ProteinNPT and other baselines. Notin et al. (2023b) report best results with a 'hybrid' uncertainty quantification strategy, however this strategy is not implemented in the publicly available code.

### A.7. Upper confidence bound acquisition function

For all ensembles, we use a parameter of $\lambda = 0.1$ within the upper confidence bound acquisition function $a(x, \lambda) = \mu(x) + \lambda\sigma(x)$, where $\mu(x)$ and $\sigma(x)$ respectively are the mean and standard deviation of the predictions of the model given an input sequence $x$.

## A.8. Implementation of ranking loss

We implement a pairwise ranking loss by using binary cross-entropy to train the model on ranking tasks by comparing predicted scores for all pairs in a batch. It computes the difference in predictions for every pair, determines the ground-truth ranking based on actual values, and applies BCE loss to these pairwise comparisons, excluding self-comparisons via a diagonal mask. An example PyTorch implementation can be found below:

```python
import torch
import torch.nn.functional as F

def full_ranking_bce(preds: torch.Tensor, targets: torch.Tensor) -> torch.Tensor:
    # Calculate pairwise differences between all predictions
    pairwise_diffs = preds[:, None] - preds[None, :]

    # Determine if each target is greater than others in a pairwise manner
    target_comparisons = targets[:, None] > targets[None, :]

    # Compute binary cross-entropy loss for pairwise comparisons
    ranking_loss = F.binary_cross_entropy_with_logits(pairwise_diffs, target_comparisons.
        float(), reduction='none')

    # Create a mask to exclude diagonal elements (self-comparisons)
    batch_size = preds.size(0)
    diag_mask = 1 - torch.eye(batch_size, device=preds.device)

    # Apply the mask and calculate the mean loss, excluding the diagonal
    masked_loss = 0.5 * ranking_loss * diag_mask
    return masked_loss.mean((-1, -2))
```

# B. Appendix: Additional Results

## B.1. Fitness Prediction: Score function and Loss Ablation

A core contribution of our work is the application of ranking-based loss function to directly fine-tune the likelihoods of masked (ESM-1v and ESM-2) and conditional autoregressive (PoET) PLM models. As an additional ablation study, we apply i) the ranking loss function in Equation (2) to the setting where a linear regression head is applied to the model embeddings, and ii) an MSE loss function is applied to the likelihood scoring functions. Below is the full table of results for each of the four settings.

Whilst we notice that the ranking loss function applied to the regression setting performs quite well, the MSE loss directly applied to the likelihood scoring functions does not.

*Table 6.* Low-n fitness prediction comparing masked and family-based PLM scoring strategies and loss functions on the Spearman correlation (higher is better). Evaluated on 8 single mutant landscapes and 5 multiple mutant landscapes from ProteinGym. ProteinNPT models use a frozen base model to produce embeddings, the base model type is provided in parentheses.

| Model Name | Scoring function | Loss | single-mutants | | | multi-mutants | | |
| --- | --- | --- | --- | --- | --- | --- | --- | --- |
| | | | $n = 32$ | $n = 128$ | $n = 512$ | $n = 32$ | $n = 128$ | $n = 512$ |
| ESM-1v (650M) | wt-marginals | mse | 0.301 | 0.282 | 0.495 | 0.446 | 0.414 | 0.544 |
| | | **ranking** | **0.479** | **0.552** | **0.641** | **0.577** | 0.642 | 0.753 |
| | linear head | mse | 0.263 | 0.415 | 0.535 | 0.494 | 0.637 | 0.771 |
| | | ranking | 0.326 | 0.437 | 0.590 | 0.474 | **0.645** | **0.777** |
| ESM-2 (650M) | wt-marginals | mse | 0.330 | 0.267 | 0.475 | 0.461 | 0.407 | 0.522 |
| | | **ranking** | **0.455** | **0.530** | **0.627** | **0.593** | **0.651** | 0.758 |
| | linear head | mse | 0.280 | 0.398 | 0.535 | 0.427 | 0.596 | 0.743 |
| | | ranking | 0.307 | 0.411 | 0.563 | 0.447 | 0.648 | **0.773** |
| PoET | likelihood | mse | 0.409 | 0.378 | 0.230 | 0.601 | 0.583 | 0.433 |
| | | **ranking** | **0.513** | **0.591** | **0.672** | **0.667** | **0.737** | **0.806** |
| | linear head | mse | 0.443 | 0.553 | 0.646 | 0.571 | 0.714 | 0.793 |
| | | ranking | 0.471 | 0.577 | 0.665 | 0.578 | 0.726 | 0.802 |

## B.2. Fitness Prediction: ProteinNPT Baseline methods

Similarly, we ablate fine-tuning the ProteinNPT model, with both ESM-1v or MSAT frozen embeddings, using the MSE loss, as proposed in (Notin et al., 2023b), and also with the ranking loss function in Equation (2). We see that the ranking loss function improves the results across all the single mutant landscapes, but results are mixed for the multi-mutant landscapes.

*Table 7.* ProteinNPT baselines (Notin et al., 2023b) that utilize frozen embeddings. Spearman correlation (higher is better) on 8 single mutant landscapes and 5 multiple mutant landscapes from ProteinGym.

| Model Name | Frozen Emb. | Loss Type | single-mutants | | | multi-mutants | | |
| --- | --- | --- | --- | --- | --- | --- | --- | --- |
| | | | $n = 32$ | $n = 128$ | $n = 512$ | $n = 32$ | $n = 128$ | $n = 512$ |
| ProteinNPT | MSAT | mse | 0.415 | 0.533 | 0.637 | 0.517 | 0.692 | **0.791** |
| | | ranking | 0.444 | **0.548** | **0.648** | 0.515 | 0.680 | 0.789 |
| | ESM-1v | mse | 0.410 | 0.497 | 0.607 | 0.438 | 0.645 | 0.769 |
| | | ranking | 0.442 | 0.527 | 0.619 | 0.435 | 0.648 | 0.780 |
| Emb. aug. | MSAT | mse | 0.424 | 0.507 | 0.553 | 0.581 | **0.696** | 0.764 |
| | ESM-1v | mse | 0.451 | 0.505 | 0.550 | 0.440 | 0.624 | 0.702 |
| OHE aug. | MSAT | mse | 0.429 | 0.467 | 0.496 | **0.616** | 0.684 | 0.763 |
| | ESM-1v | mse | **0.466** | 0.502 | 0.526 | 0.460 | 0.566 | 0.711 |
| OHE | – | mse | 0.144 | 0.314 | 0.488 | 0.268 | 0.473 | 0.664 |

## B.3. Additional Results: Generalisation of Seen vs Unseen residue positions

We provide Spearman correlation results for the $n = 128$ fitness prediction setting specifically looking at out-of-distribution generalisation at unseen mutated residues. That is, for single mutant landscapes the number of *seen* mutant positions in the training datasets (varies per landscape): min=60, max=114, mean=82.67, and the number of *unseen* mutant positions: min=17, max=657, mean=172.46. For single-mutant landscapes, we classify test set sequences as seen (mutation present in the train set) or unseen (mutation absent). For multi-mutant landscapes, a test set sequence is considered seen if it has up to two mutations that occur in the train set sequences; otherwise, it is unseen. The number of seen sequences in the test set of multi-mutant landscapes is on average 2,124 (equivalent to approximately 42% of sequences in the test set) and 2,875 unseen sequences. Note, the test set size varies per protein landscape, however, for multi-mutant landscapes we limit it to 5,000 sequences.

*Table 8.* Seen vs Unseen Spearman correlation scores (higher is better) evaluated on the 8 single mutant landscapes (left) and 5 multi-mutant landscapes (right) for the $n = 128$ dataset setting.

| Model Name | Scoring Function | Loss Type | single-mutants | | multi-mutants | |
| --- | --- | --- | --- | --- | --- | --- |
| | | | Seen | Unseen | Seen | Unseen |
| ESM-1v (650M) | linear head | mse | 0.460 | 0.331 | 0.646 | 0.604 |
| | | ranking | 0.492 | 0.315 | 0.651 | 0.609 |
| | wt-marginals | mse | 0.350 | 0.234 | 0.412 | 0.387 |
| | | ranking | **0.592** | **0.509** | 0.652 | 0.643 |
| ESM-2 (650M) | linear head | mse | 0.453 | 0.297 | 0.605 | 0.556 |
| | | ranking | 0.447 | 0.335 | 0.649 | **0.622** |
| | wt-marginals | mse | 0.329 | 0.192 | 0.423 | 0.385 |
| | | ranking | **0.568** | **0.455** | **0.658** | 0.620 |
| PoET | linear head | mse | 0.571 | 0.517 | 0.700 | 0.695 |
| | | ranking | 0.601 | 0.535 | 0.716 | 0.715 |
| | likelihood | mse | 0.382 | 0.366 | 0.576 | 0.613 |
| | | ranking | **0.612** | **0.549** | **0.728** | **0.741** |
| ProteinNPT (MSAT) | - | mse | 0.563 | 0.462 | 0.694 | 0.670 |
| | - | ranking | 0.579 | 0.474 | 0.675 | 0.664 |
| ProteinNPT (ESM-1v) | - | mse | 0.529 | 0.420 | 0.641 | 0.601 |
| | - | ranking | 0.553 | 0.465 | 0.642 | 0.607 |

## B.4. Additional Results: More Expressive Masked Scoring Functions

In Table 9 we show the complete results for additional masked PLM scoring functions that attempt to capture the epistasis effects in the multi-mutant landscapes. We provide results for ESM-1v and ESM-2 for the additional strategies applied to five multi-mutant landscapes, as introduced in Meier et al. (2021) and Johnson et al. (2024).

Note, due to GPU memory requirement of the masked-modulo strategy, we reduce the hyperparameters relative to the length of the protein sequence in order to fit on an H100 GPU. For example, batch size ($B$) and $K$ for each landscape were set to GRB: ($B$=8, $K$=8), GFP: ($B$=4, $K$=8), HIS: ($B$=8, $K$=8), PABP: ($B$=4, $K$=4) and CAP: ($B$=2, $K$=7).

For each strategy proposed in Meier et al. (2021), as an ablation, we modify them to consider the likelihood of every token in the sequence when computing the score, rather than just the likelihood at the mutations (we denote these modified strategies with $'$). As a result, the summation in Equation (5) is modified from $\sum_{i:x_i^{\text{wt}} \neq x_i}$ to $\sum_i$.

*Table 9.* Masked PLM scoring strategies evaluated on five multi-mutant ProteinGym landscapes, where $M$ is the number of mutations in a sequence, $B$ is the batch size, and $K$ is the masked-modulo constant (set to 4 or 8 depending on landscape).

| Scoring Function | Steps | Loss | ESM-1v | | | ESM-2 | | |
|---|---|---|---|---|---|---|---|---|
| | | | $n = 32$ | $n = 128$ | $n = 512$ | $n = 32$ | $n = 128$ | $n = 512$ |
| wt-marginals | 1 | mse | 0.446 | 0.414 | 0.544 | 0.461 | 0.407 | 0.522 |
| (Meier et al., 2021) | | ranking | 0.577 | 0.642 | 0.753 | **0.593** | **0.651** | 0.758 |
| masked-mt-marginals | $B$ | mse | 0.389 | 0.362 | 0.562 | 0.392 | 0.328 | 0.561 |
| (Meier et al., 2021) | | ranking | 0.522 | 0.650 | 0.755 | 0.559 | 0.651 | 0.766 |
| masked-mt-marginals′ | $B$ | mse | 0.572 | 0.604 | 0.602 | 0.571 | 0.629 | 0.632 |
| | | ranking | 0.555 | 0.586 | 0.647 | 0.558 | 0.606 | 0.635 |
| mt-marginals | $B$ | mse | 0.214 | 0.291 | 0.533 | 0.264 | 0.330 | 0.530 |
| (Meier et al., 2021) | | ranking | 0.351 | 0.578 | 0.754 | 0.398 | 0.591 | 0.750 |
| mt-marginals′ | $B$ | mse | **0.579** | 0.617 | 0.622 | 0.565 | 0.612 | 0.627 |
| | | ranking | 0.552 | 0.644 | 0.767 | 0.525 | 0.646 | **0.771** |
| masked-modulo | $K \cdot B$ | mse | **0.579** | 0.621 | 0.645 | 0.568 | 0.586 | 0.620 |
| (Johnson et al., 2024) | | ranking | 0.529 | **0.654** | **0.769** | 0.534 | **0.661** | **0.774** |

## B.5. Compute requirements

All experiments were run on either A100 or H100 NVIDIA GPUs. Compute required for a single fine-tuning run varies based on the model, the length of the protein sequences, and the size of the dataset. We provide representative timings averaged over the 8 single mutant landscapes for $n = 512$ in Table 10. Design experiments involved 10 rounds of fine-tuning and therefore required roughly ten times the computation of a single fine-tuning run.

*Table 10.* Representative run times for fine-tuning ($n = 512$) averaged over 8 single-mutant landscapes and across 3 seeds, on an H100 GPU.

| Model name | Time |
|---|---|
| ProteinNPT (MSAT) | 24 m |
| ProteinNPT (ESM-1v) | 34 m |
| ESM-1v (linear head, mse) | 35 m |
| ESM-1v (wt-marginals, rank) | 7 m |
| ESM-2 (linear head, mse) | 15 m |
| ESM-2 (wt-marginals, rank) | 4 m |
| PoET (linear head, mse) | 7 m |
| PoET (likelihood, rank) | 7 m |

## B.6. Zero-shot PLM performance

As discussed in Section 5.2 Result 2, the ranking-based fine-tuning performance of PoET is not attributed directly to higher zero-shot performance of the base PLM. We evaluate the zero-shot performance of the base models here, on the single mutant landscapes using the $n = 128$ test split and report the Spearman correlation between likelihood scoring function and the fitness measurement. The MSAT zero-shot predictions are taken from Notin et al. (2023a) for our test splits.

*Table 11.* Zero-shot Spearman correlation on the $n = 128$ test splits for the base PLM models.

| Base Model | Zero-shot Spearman |
|---|---|
| MSA Transformer (MSAT) | 0.399 |
| ESM-1v (wt-marginals, rank) | 0.437 |
| ESM-2 (wt-marginals, rank) | 0.372 |
| PoET (likelihood, rank) | 0.417 |

## B.7. Low-n Fitness Prediction with Ensemble Models

*Table 12.* Low-n fitness prediction Spearman results comparing the masked- and family-based MSA-ensemble models to their single model counterparts. Averaged over three seeds and 8 single mutant landscapes (left) and five multi-mutant landscapes (right).

| Model Name | Scoring Fn. | Loss | Single-mutants | | Multi-mutant | |
|---|---|---|---|---|---|---|
| | | | $n = 32$ | $n = 128$ | $n = 32$ | $n = 128$ |
| PNPT (MSAT) | - | mse | 0.420 | 0.532 | 0.511 | 0.696 |
| PNPT (MSAT) w/ dropout | - | mse | 0.421 | 0.532 | 0.512 | 0.696 |
| ESM-2 | wt-marginal | ranking | 0.455 | 0.530 | 0.593 | 0.651 |
| | | mse | 0.330 | 0.267 | 0.461 | 0.407 |
| | linear head | ranking | 0.307 | 0.411 | 0.447 | 0.648 |
| | | mse | 0.280 | 0.398 | 0.427 | 0.596 |
| ESM-2 ensemble | wt-marginal | **ranking** | **0.477** | **0.553** | **0.621** | **0.683** |
| | | mse | 0.347 | 0.335 | 0.507 | 0.440 |
| | linear head | ranking | 0.357 | 0.435 | 0.511 | 0.694 |
| | | mse | 0.342 | 0.428 | 0.505 | 0.658 |
| PoET | likelihood | ranking | 0.514 | 0.594 | 0.667 | 0.736 |
| | | mse | 0.409 | 0.378 | 0.601 | 0.583 |
| | linear head | ranking | 0.468 | 0.575 | 0.574 | 0.723 |
| | | mse | 0.445 | 0.554 | 0.582 | 0.715 |
| PoET MSA-ensemble | likelihood | **ranking** | **0.524** | **0.607** | **0.696** | **0.757** |
| | | mse | 0.412 | 0.390 | 0.623 | 0.609 |
| | linear head | ranking | 0.504 | 0.598 | 0.618 | 0.744 |
| | | mse | 0.486 | 0.591 | 0.632 | 0.736 |

## B.8. Additional sequence design recall curves

### B.8.1. ESM-2 MASKED-ENSEMBLES

ESM-2 masked-ensembles comparing the wt-marginal scoring strategy fine-tuning via ranking loss to the linear regression head fine-tuned with MSE loss. AUC = area under the curve (higher is better). Each ensemble contains 5 members, with more details specified in Appendix A.4. Evaluated on 8 single mutant landscapes (left) and 5 multiple mutant landscapes (right).

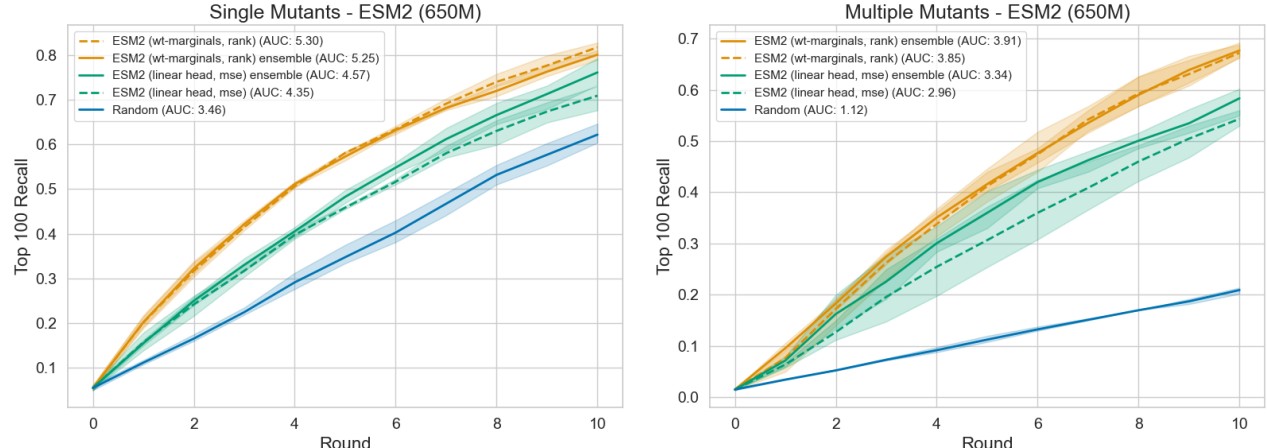

*Figure 2.* ESM-2 (650) masked-ensembles (left): single mutation landscapes. (right) multiple mutation landscapes.

### B.8.2. FAMILY-BASED PLMS

PoET MSA-ensemble comparing the likelihood fine-tuning via ranking loss to the linear regression head fine-tuned with MSE loss. AUC = area under the curve (higher is better). Each ensemble contains 5 members, with more details specified in Appendix A.5. Evaluated on 8 single mutant landscapes (left) and 5 multiple mutant landscapes (right).

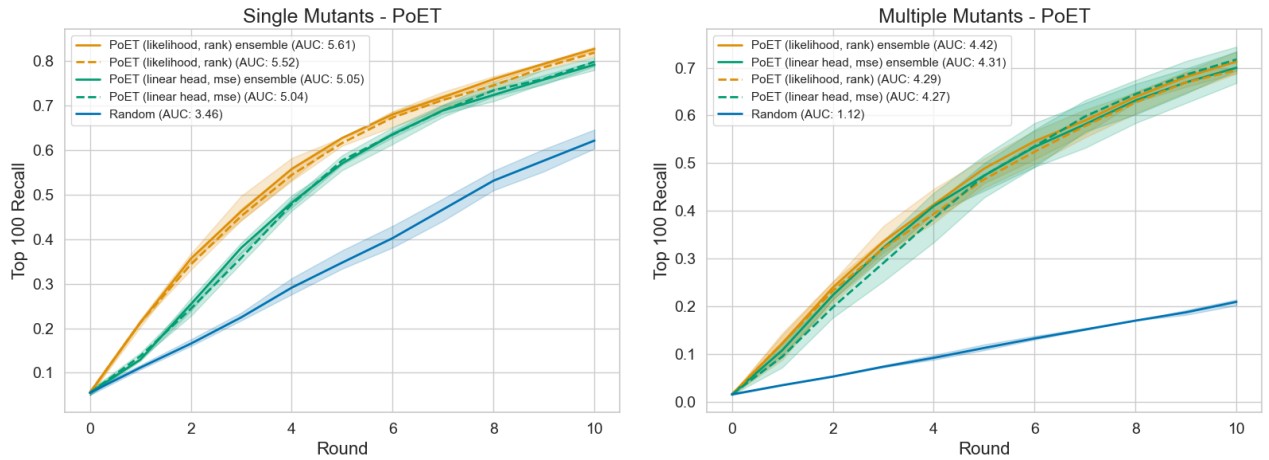

*Figure 3.* PoET MSA-ensembles (left): single mutants landscapes. (right) multiple mutant landscapes.

### B.8.3. PROTEINNPT BASELINES

ProteinNPT baseline methods taken from (Notin et al., 2023b). AUC = area under the curve (higher is better). Uncertainty is calculated using MC dropout, with more details specified in Appendix A.6. Evaluated on 8 single mutant landscapes (left) and 5 multiple mutant landscapes (right).

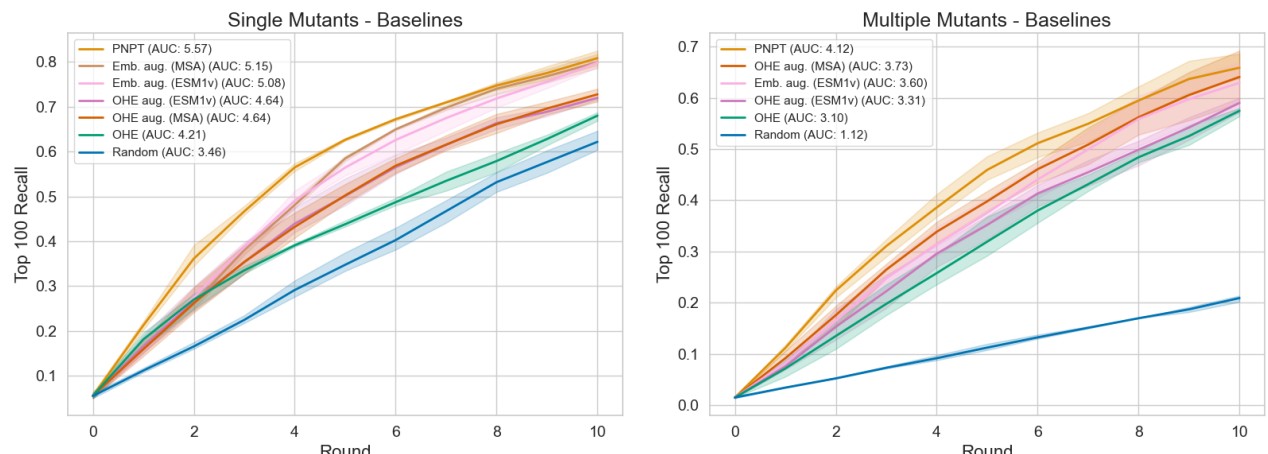

*Figure 4.* ProteinNPT baselines (left): single mutation landscapes. (right) multiple mutation landscapes.

### B.8.4. SINGLE MUTANT LANDSCAPE RESULTS

Each method is evaluated on each of the 8 single mutation landscapes and each of the 5 multiple mutation landscape, repeated across 3 random seeds.

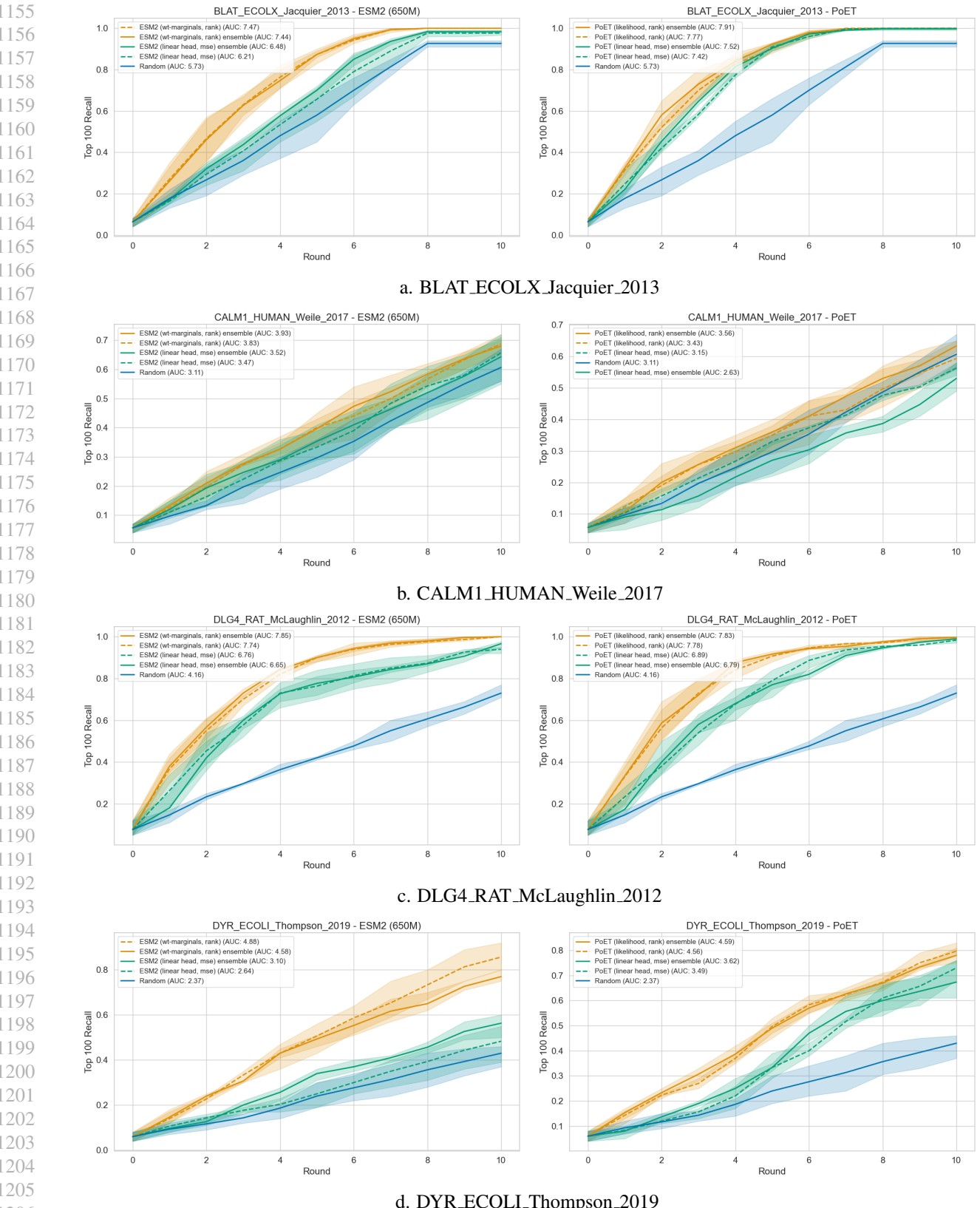

*Figure 5.* Sequence design top 100 sequences recall results on each of the single mutation landscapes. Masked PLM ESM-2 and ESM-2 masked-ensembles (left), Family-based PoET and PoET MSA-ensemble (right).

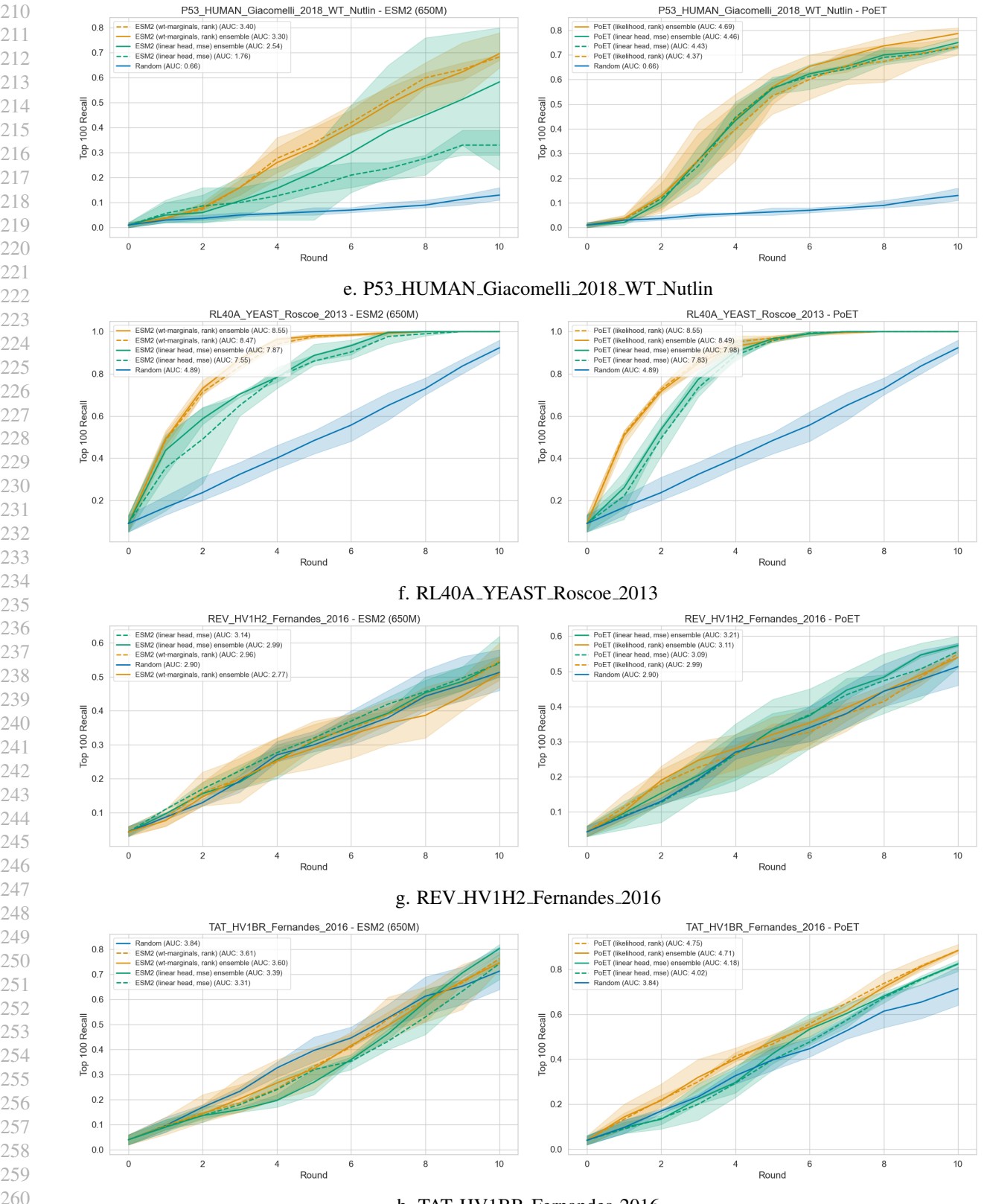

e. P53_HUMAN_Giacomelli_2018_WT_Nutlin

f. RL40A_YEAST_Roscoe_2013

g. REV_HV1H2_Fernandes_2016

h. TAT_HV1BR_Fernandes_2016

*Figure 6.* Sequence design top 100 sequences recall results on each of the single mutation landscapes. Masked PLM ESM-2 and ESM-2 masked-ensembles (left), Family-based PoET and PoET MSA-ensemble (right).

### B.8.5. MULTIPLE MUTANT LANDSCAPE RESULTS

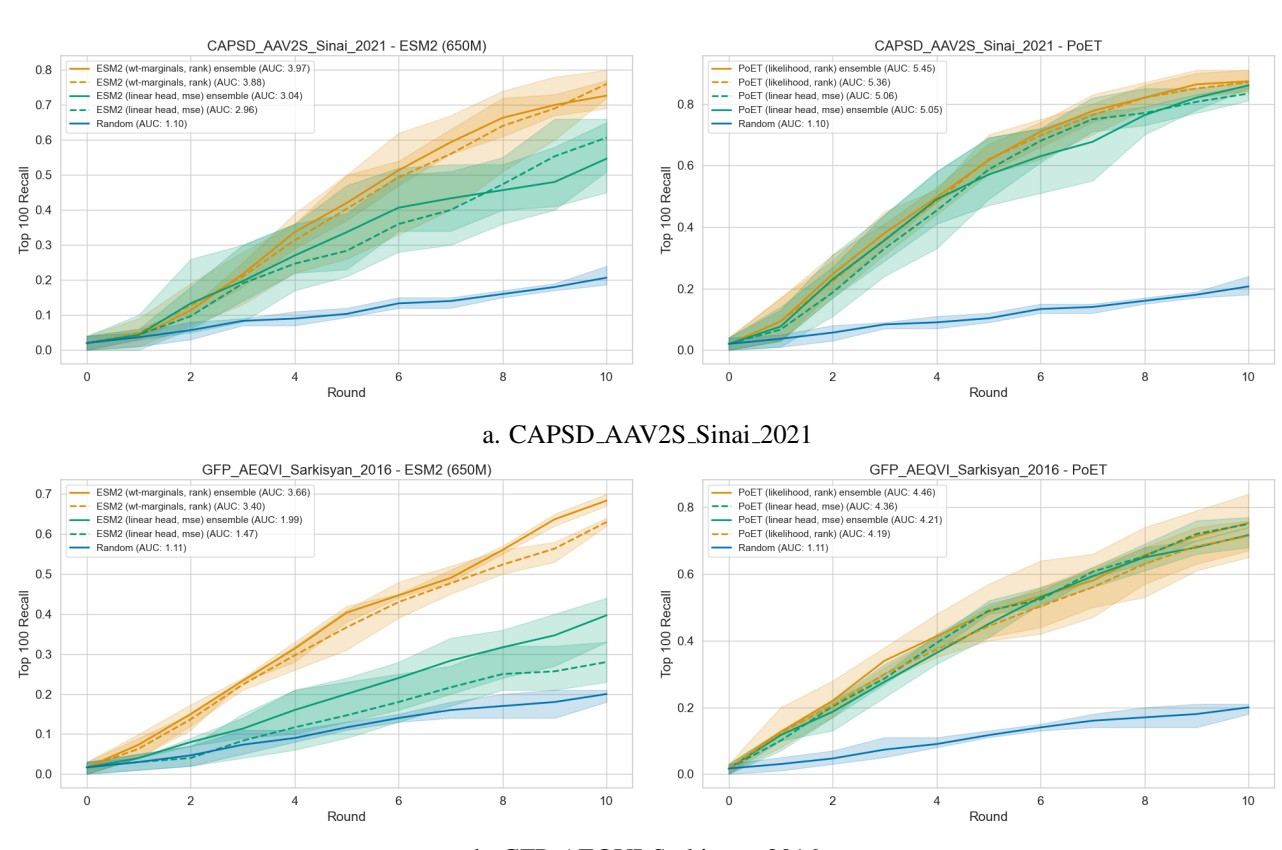

a. CAPSD_AAV2S_Sinai_2021

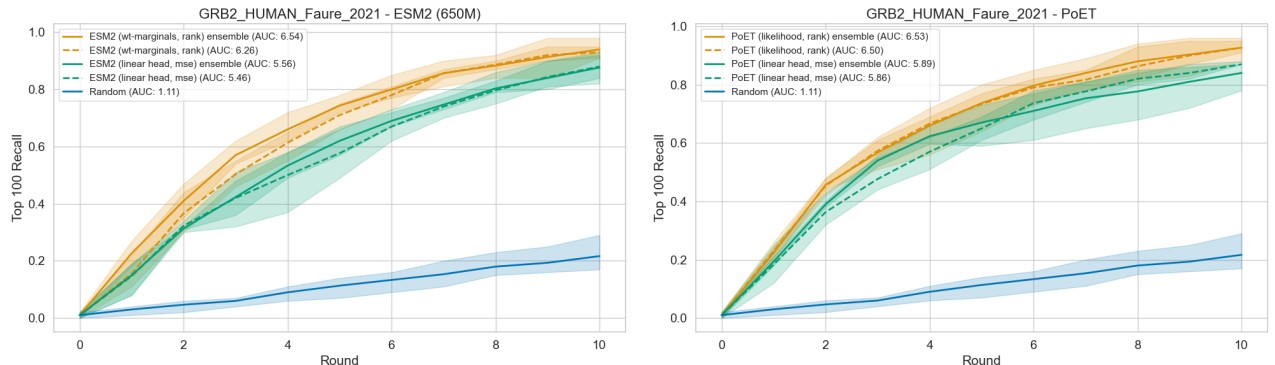

b. GFP_AEQVI_Sarkisyan_2016

c. GRB2_HUMAN_Faure_2021

*Figure 7.* Sequence design top 100 sequences recall results on each of the multiple mutation landscapes. Masked PLM ESM-2 and ESM-2 masked-ensembles (left), Family-based PoET and PoET MSA-ensemble (right).

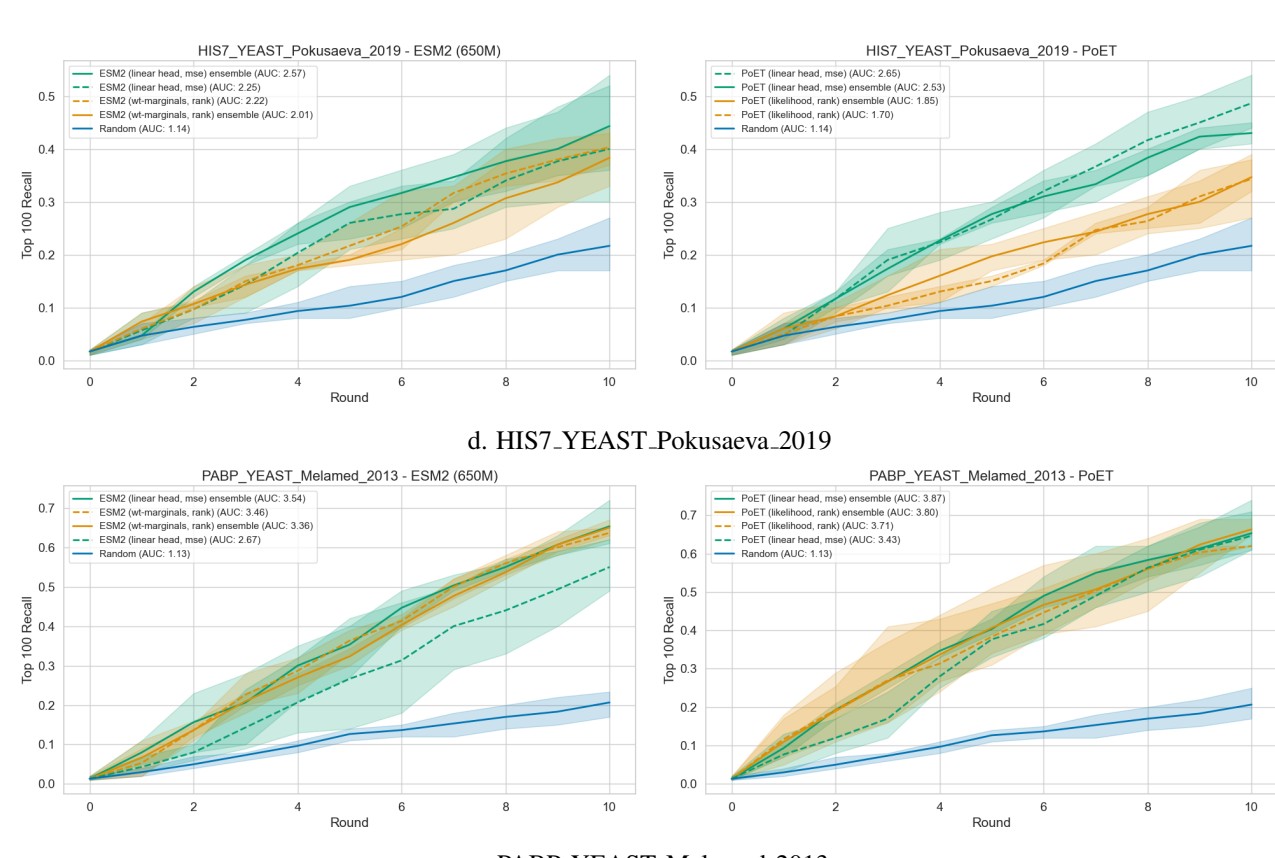

d. HIS7_YEAST_Pokusaeva_2019

e. PABP_YEAST_Melamed_2013

*Figure 8.* Sequence design top 100 sequences recall results on each of the single mutation landscapes. Masked PLM ESM-2 and ESM-2 masked-ensembles (left), Family-based PoET and PoET MSA-ensemble (right).

*Table 13.* Sequence design full table of AUC results, across models, scoring functions, loss functions, ensembles, and baseline methods. AUC = area under the curve (higher is beter) presented for both Top 30% recall (as per Notin et al. (2023b)), and recall over the Top 100 sequences in the pool. Averaged over 8 single mutant landscapes (left) and 5 multiple mutant landscapes (right).

| Model Name | Scoring Fn. | Loss | Single-mutants | | Multi-mutant | |
|---|---|---|---|---|---|---|
| | | | Top 100 Recall | Top 30% Recall | Top 100 Recall | Top 30% Recall |
| ESM-2 (650M) | linear head | mse | 2.145 | 2.961 | 4.402 | 4.354 |
| | | ranking | 2.372 | 3.505 | 4.614 | 4.873 |
| | wt-marginals | mse | 2.005 | 2.786 | 4.455 | 4.521 |
| | | ranking | 2.579 | 3.845 | 5.031 | 5.296 |
| ESM-2 (650M) ensemble | linear head | mse | 2.309 | 3.339 | 4.595 | 4.569 |
| | wt-marginals | ranking | 2.590 | 3.907 | 5.125 | 5.252 |
| PoET | linear head | mse | 2.683 | 4.271 | 4.967 | 5.041 |
| | | ranking | 2.732 | 4.330 | 5.209 | 5.532 |
| | likelihood | mse | 1.769 | 1.728 | 3.881 | 3.802 |
| | | ranking | 2.764 | 4.289 | 5.212 | 5.524 |
| PoET ensemble | linear head | mse | 2.776 | 4.312 | 5.057 | 5.049 |
| | | ranking | 2.775 | 4.408 | 5.260 | 5.552 |
| | likelihood | mse | 1.767 | 1.726 | 3.741 | 3.765 |
| | | ranking | 2.797 | 4.418 | 5.266 | 5.612 |
| PNPT (MSA) w/ dropout | - | mse | 2.604 | 4.121 | 5.099 | 5.567 |
| Emb. aug. (ESM-1v) | - | mse | 2.617 | 3.603 | 4.947 | 5.076 |
| OHE | - | mse | 2.326 | 3.098 | 4.421 | 4.211 |
| OHE aug. (MSA) | - | mse | 2.680 | 3.734 | 4.724 | 4.641 |
| OHE aug. (ESM-1v) | - | mse | 2.463 | 3.311 | 4.772 | 4.644 |

