# OpenReview forum: "Likelihood-based Finetuning of Protein Language Models for Few-shot Fitness Prediction and Design"
_ICML.cc/2025/Conference — Submitted to ICML 2025_

### Official Review · Reviewer_QopU · 2025-02-17

**Overall Recommendation:** 4

**Summary:**

The authors want to use pre-trained protein language models for supervised prediction. Rather than classical fine-tuning to maximize regression accuracy with a linear probe, these authors suggest fine tuning the ordering of the likelihoods, which are good zero-shot predictors. Predictably, this method works well in the low-N regimen. They validate on proteingym.

**Claims And Evidence:**

sure

**Essential References Not Discussed:**

No

**Experimental Designs Or Analyses:**

Yes.

**Methods And Evaluation Criteria:**

sure

PLMs are neat, but you should also consider other methods such as Kermut, which, as I understand, is currently state of the art -- https://arxiv.org/abs/2407.00002. HAving the zero-shot numbers in the tables would also be useful.

**Other Comments Or Suggestions:**

.

**Other Strengths And Weaknesses:**

Very straightforward approach (strength). I tried this approach 4 years ago on VAEs with a different loss and, of course, it worked. [This paper](https://arxiv.org/pdf/2412.07763) also has a similar paradigm of fitting the likelihoods in low-N iterative design (with a very very different methodology), and it works. It's surprising to me that the authors applied this to large PLMs with the BT loss and saw improvements with n as high as 512.

Few things that would improve the paper:
1. Error bars in spearmans,
2. Discussion of how to do the fine-tuning: learning rates, early stopping etc... Ideally, a sensitivity plot.
3. Can you write the objective as the likelihood of a generative process? For example, the Pearson correlation is the marginal likelihood if you assume a linear relation between liks and labels with a improper uniform prior. Since the paper is so simple, maybe it wouldn't be too much to ask the authors to try different losses, the Pearson correlation for example.

**Questions For Authors:**

.

**Relation To Broader Scientific Literature:**

Fine.

**Theoretical Claims:**

Sure

---

> ### Author Rebuttal · Authors · 2025-03-31
>
> Thank you very much for your feedback - we appreciate the review. Thank you for acknowledging the simplicity and applicability of our fine-tuning approach as a strength, whilst other reviewers discounted those same traits. Responses to your comments are given below. If we have sufficiently addressed your concerns, we would kindly ask that you champion the paper during the reviewer discussion phase, or accordingly, please let us know if further clarification is needed.
>
> Relationship to Kermut:
>
> We thank the reviewers for highlighting Kermut! Kermut does indeed outperform ProteinNPT on the ProteinGym suite of fitness prediction tasks. There are a few reasons why we did not consider it as a fair baseline in our work:
>
> 1) Kermut introduces a novel “biologically meaningful” composite kernel function with which to compare two given proteins. However, this composite kernel utilizes vastly more information than our method (and the collection of baselines we ultimately chose) - in particular, they make use of an inverse folding model that provides structural information to three of their four kernel components. Whilst we agree with the reviewer that this is a neat approach, and one that would likely improve results, including structural information is orthogonal to our work and we instead solely focus on the valuable task of fine-tuning sequence-based PLMs.
>
> 2) Kermut is not SOTA without structural information. I.e. Their ablation demonstrates that the sequence-only kernel, with representations from ESM-2, is outperformed (in Spearman random data setting) by ProteinNPT (0.744 - 0.033 = 0.711 versus 0.73). We directly compare to, and outperform, ProteinNPT.
>
> 3) Ultimately, we see Kermut’s primary contribution as “how to incorporate the information contained in PLMs into GPs (e.g. to provide meaningful uncertainty estimates)”, an important research topic in its own right. However, we argue that this is orthogonal to our work, and potentially could be combined in future work.
> On this point, we have added further discussion to our paper making explicit these points.
>
>
> Observed Trends:
>
> As the reviewer correctly points out, it is somewhat surprising that the proposed ranking approach is still effective in some  landscapes at N=512. We see a clear trend that as N increases, the performance gap of the proposed likelihood ranking approach shrinks relative to the standard parametric regression approach.
>
> To support this claim, and the main claims made in the paper, please find additional fitness prediction results for N=24 and N=48 in the Reviewer vxYr (Experimental Analysis) section.
>
>
> We will include per-landscape results in the Appendix, and also Error bars in the Spearman results.
>
> Zero-shot values for base PLMs are provided in Table 11 in Appendix B.
>
>
> Objective as the likelihood of a generative process:
>
> Please could you expand upon how Pearson correlation could be used as a training loss?

---

> > ### Comment · Reviewer_QopU · 2025-04-01
> >
> > My concerns are addressed. I think this is a simple method that takes the idea of using evolutionary likelihoods as zero-shot predictors and extrapolates it to the multi-shot setting. I have implemented such a method before and seen its success, but I don't think this has been published anywhere.
> >
> > Skimming through the other reviews, it seems that the authors are being asked to demonstrate that their method works in every case against the enormous amount of baselines now popular for protein design. I think this doesn't recognize how simple the author's method is: many groups are already using likelihoods for zero-shot prediction -- these authors just ask them to train these likelihoods on the few measurements they have. I think this work functions as a super simple drop-in baseline that miraculously hasn't been reported, to my own frustration.
> >
> > It also seems that the other reviewers are confusing this work with superficially similar works, for example Contrastive losses as generalized models of global epistasis by Brookes et al., 2024 (nothing in common except proteins and a Bradley Terry loss are involved; calling the Brookes paper more "theoretical" is also a little strange -- I in fact recommended this paper for rejection because of its glib theory when I reviewed it a few years ago).
> >
> > Regarding fitting to Pearson correlation, this is what the authors of [this paper do](https://arxiv.org/pdf/2412.07763), although it's in-context rather than fine-tuned. I'm just saying you can swap minimizing the BT loss for maximizing the Spearman correlation.

---

### Official Review · Reviewer_J3GB · 2025-03-06

**Overall Recommendation:** 2

**Summary:**

This paper extends a ranking-based fine-tuning strategy to various protein language models, including masked PLMs and autoregressive family-based PLMs. Specifically, it introduces different scoring functions for these models and uses conditional ranking loss to fine-tune them. The experiments on fitness prediction and sequence design show that ranking-based fine-tuning outperforms supervised fine-tuning.

**Claims And Evidence:**

The paper claims that ranking-based finetuning outperforms supervised finetuning in fitness modeling, as shown in the experiment. However, it's unclear if this claim also applies to auto-regressive models like esm and ProtGPT.

**Essential References Not Discussed:**

1. I encourage the author to discuss ranking-based loss (DPO) used in protein design more (see prior part).
2. I think the search-based model-guided protein sequence design method should be included and studied, e.g., AdaLead, PEX, LatProtRL [1,2,3].

References:
[1] AdaLead: A simple and robust adaptive greedy search algorithm for sequence design
[2] Proximal Exploration for Model-guided Protein Sequence Design
[3] Robust Optimization in Protein Fitness Landscapes Using Reinforcement Learning in Latent Space

**Experimental Designs Or Analyses:**

The whole experiment part (fitness prediction and sequence design) makes sense to support this method, however, essential datasets and baselines are missing, which may hide the potential use of such a method.
1. In the fitness prediction, this paper only shows a subset of the ProteinGym dataset results. Can you provide more results on ProteinGym, as it's a gold standard for fitness prediction?
2. In the sequence design task, there are a lot of baselines using directed evolution (searching-based method, e.g. AdaLead, PEX...). I wonder if the fine-tuned PLMs can be used as landscapes for these methods to improve their performance.
3. Also, there are many different datasets in sequence design task (see AdaLead and PEX paper), could you provide results on these datasets?
4. Auto-regressive PLMs should also be considered as baselines in these tasks.
5. Please provide more details of experiments, especially for sequence design task.

**Methods And Evaluation Criteria:**

The method in this paper makes sense, as we can use an alternative way to fine-tune protein language models beyond regression-based finetuning.

**Other Comments Or Suggestions:**

No.

**Other Strengths And Weaknesses:**

Strengths:
1. The empirical results in section 5 (fitness prediction) show some valuable suggestions when fine-tuning PLMs.

Weaknesses:
1. The writing is kind of unclear and confused, for example, the second last part of introduction and the last paragraph of introduction seem redundant, I encourage the author to refine the writing flow, and maybe explainable figures can be used to help understand.
2. The method seems to be a natural extension from ranking loss by adjusting different score functions, which might not be quite interesting.
3. The experiment lacks some baselines and results.

**Questions For Authors:**

Please see other parts.

**Relation To Broader Scientific Literature:**

The idea of using ranking-based loss (I believe it's DPO) has been studied in protein prediction/design communities [1,2,3], this method only extends that for mask PLMs and family-based autoregressive PLMs by adjusting score functions (Eq 5,6).

References:
[1] Aligning protein generative models with experimental fitness via Direct Preference Optimization
[2] Fine-tuning protein Language Models by ranking protein fitness
[3] Antigen-specific antibody design via direct energy-based preference optimization

**Theoretical Claims:**

The math formulations about score functions for different PLMs seem correct and make sense.

---

> ### Author Rebuttal · Authors · 2025-03-31
>
> Thank you very much for your review. Responses to your comments are given below. However, we highlight a number of misunderstanding and factual errors in the provided review. We kindly request that you re-evaluate your review in light of these clarifications. If we have sufficiently addressed your concerns, we kindly ask that you raise your score accordingly, or let us know if further clarification is needed.
>
> One fundamental misunderstanding repeated multiple times is that “The paper claims that ranking-based finetuning outperforms supervised finetuning in fitness modeling, as shown in the experiment.”
>
> To clarify, our proposed approach directly adapts PLM likelihoods via a ranking-based loss function and is in fact still a supervised fine-tuning approach. Both settings, regression and ranking, utilize a supervised learning paradigm via ground truth fitness scores in a ProteinGym DMS. Our results demonstrate that by directly applying ranking losses to the PLM likelihoods we can better utilize limited labelled data.
>
> Furthermore, the two comments “... However, it's unclear if this claim also applies to auto-regressive models like esm and ProtGPT.” and “Auto-regressive PLMs should also be considered as baselines in these tasks.”
>
> Contains a misunderstanding of the models used in our work. Our paper introduces fine-tining methods specifically for conditional autoregressive models, like the family-based PoET  in Section 4.1 “Conditional scoring functions”, and for **masked** language models ESM-1v and ESM-2 in Section 4.2 “Scoring functions for masked PLMs. That is, ESM is not an auto-regressive PLM as the reviewer suggests.
>
> Another misunderstanding is that our work targets open-ended optimization via search. In the final paragraph of our Introduction, we state “...we study the modelling problem in silico which mimics ground truth values being available from wet lab experiments.”
>
> We agree with the reviewer that search-based methods are an important field in proteinomics, however, it is not the focus of our work. Therefore, we respectfully refute the reviewer’s claim that methods such as directed evolution, PEX, AdaLead and LatProtRL [1,2,3] are “essential references not discussed”, and “should be included and studied”.
>
> Leveraging the large collection of datasets available in ProteinGym allows for more concise evaluation of the proposed methodologies. This is a common approach taken in ProteinNPT (Notin, P. et. al. NeurIPS 2023), PoET (Truong Jr & Bepler, NeurIPS 2023), Kermut (Groth, P.M. et al. NeurIPS 2024) and Metalic (Beck, J. et. al. ICLR 2025). Conversely, it is well known that creating a biological meaningful oracle model to provide feedback, as per the reviewer suggess, is arguably a more challenging task than the optimization itself [4, 5].
>
> [4] Buttenschoen, M., et. al. Posebusters: Ai-based docking methods fail to generate physically valid poses or generalise to novel sequences. Chemical Science, 2024
>
> [5] Surana, S. et. at. Overconfident Oracles: Limitations of In Silico Sequence Design Benchmarking, AI for Science Workshop at ICML 2024
>
> “...discuss DPO used in protein design more…”:
>
> We kindly thank the reviewer for their references and will expand upon our current Section 4.5. “Relationship to preference learning for LLMs” where we discuss the benefits and limitations of DPO for our setting, e.g. that the KL penalty term hinders PLM adaptation in low-data settings.
>
> In fact, we do cite work suggested [2] in this section, and in Section 2 Related Work (Page 2). Whilst it is concurrent work, they focus on a non-conditional autoregressive model, limiting their application to the family-conditioned model PoET, and therefore will not achieve our SOTA results (Table 1 and Figure 1).
>
> Furthermore, [1] introduces a structure conditioned generative LM, pre-trained using DPO. Whilst this is an important task, it is orthogonal to our task of fine-tuning **pre-trained PLMs**. Our focus (as in ProteinNPT, PoET and Metalic) is to adapt the performance of pre-trained models using limited (small-N) datasets. Our work makes no claims on the pre-training regimes.
>
> Similarly, [3] introduces a preference-based fine-tuning scheme for a pre-trained structure-conditioned diffusion model. Whilst conceptually related at a high level, their goal is de novo generation of high binding antibodies, and they incorporate feedback from a physics-based energy oracle, more similar to the open-search setting discussed above.
>
> For additional low-N fitness prediction results, please see Review vxYr (Experimental Analysis).
>
> “No supplemental material”. This is not true. Other reviewers acknowledged the Supplemental materials.
>
> Natural extension from ranking losses:
>
> In fact, we feel this is a strength of our work: it is a natural and necessary extension of the current literature, and we have demonstrated it has wide ranging applicability to different families of PLMs, as per the comments from Reviewer AKmL and QopU.

---

### Official Review · Reviewer_AKmL · 2025-03-12

**Overall Recommendation:** 3

**Summary:**

To train protein sequence to fitness regression models, it is attractive to fine tune protein language models (PLMs), as these have prior knowledge about the constraints underlying protein function, etc. The authors provide a specific fine tuning strategy, where the likelihood of a generative model is optimized using a ranking loss. They show that it works for a number of different generative models and compare to recent alternative modeling approaches.

**Claims And Evidence:**

The results on low-N protein function prediction are strong, showing an improvement over important recent baselines. However, the benchmarking is done on a small subset of the tasks available in ProteinGym and in an eval setup (low-N) that is important, but non-standard. I did not find the results for multi-round fitness optimization convincing. See below for details.

**Essential References Not Discussed:**

You should discuss Widatalla et al "Aligning protein generative models with experimental fitness via Direct Preference Optimization"

There is significant overlap with Beck 2024: "Metalic: Meta-Learning In-Context with Protein Language Models" in terms of motivation and the eval setup. Can you please explain the similarities and differences between your works?

**Experimental Designs Or Analyses:**

There is no clear quantitative comparison of methods in the multi-round optimization section. Curves are compared to each other, with no well-motivated way to compare the curves. Further, the error bars, particularly in the left figure, suggest that methods perform very similarly.

For fitness prediction, models are benchmarked in the low-N regime, where the training dataset is sampled to be very small. The baseline modeling approaches (particularly, ProteinNPT) were not developed for this use case and I worry that they were not retrained and subjected to hyper-parameter tuning for the type of evaluation task in this paper.

**Methods And Evaluation Criteria:**

The proposed method is well motivated and previously established in some non-archival workshop papers. The authors did a good job of applying it to a wide variety of pretrained models (autoregressive, masked, etc) and providing some model-specific ablations (such as different masking strategies for approximating likelihoods from a masked model).

The datasets (proteingym) are a popular benchmark, but the authors use them in a non-standard way, where models are trained on a small subset.

**Other Comments Or Suggestions:**

I found Result 1 confusing in sec 6.2, since it doesn't concern optimization. Shouldn't that appear in the previous section?

**Other Strengths And Weaknesses:**

I appreciate the eval setup where you generalize to positions with no mutations in training. This led me to believe that the models were learning something non-trivial.

**Questions For Authors:**

The evaluation focuses on the low-N regime. I assuming this because you achieved sub-optimal results on the full ProteinGym setup. What were the results? How far behind SOTA was it?

I'm curious about an 'evo tuning' baseline, which has appeared in a variety of prior papers. If you had just fine-tuned your language model on natural homologs to the wildtype for each task, how would that have performed compared to fine tuning that depends on experimental data?

multi-round eval: was it the same initial training set for each method? I'm assuming that the contents of this initial set can have a huge impact on optimization performance. What are the error bars in fig 1? Are any differences statistically significant?

"Since all sequences are scored under the residue distributions obtained by inputting the wild-type sequence to the model, a set of mutated sequences of arbitrary size can be scored using a single forward pass, making it extremely efficient." I found this confusing. Is it valid to use the model without inputting any mask tokens? doesn't it just copy the input sequence?

"As an ablation, we compare to a mean square error (MSE) loss applied to the same scoring function."
   A-priori, the log likelihood scores from the model could be wildly different from the units of the fitness scores, which could make MSE training unstable. Did you try anything basic, like scaling the likelihood scores to be in the range [0, 1], for example?


===After Authors Response===
See comment below. Thank you for the thorough discussion. I have raised my score to weak accept.

**Relation To Broader Scientific Literature:**

The relation to prior work is a bit awkward. As far as I can understand, the modeling technique has appeared in multiple previous workshop papers. The primary contribution of this paper is that it provides more comprehensive benchmarking.

**Theoretical Claims:**

Not applicable

---

> ### Author Rebuttal · Authors · 2025-03-31
>
> Thank you very much for your detailed feedback - we appreciate the review. If we have sufficiently addressed your concerns, we kindly ask that you raise your score accordingly, or let us know if further clarification is needed.
>
> Experimental Analysis
>
> Clarification: We do provide a clear quantitative comparison of methods in the multi-round optimisation setting. We report the Area Under Curve (AUC) in all design experiment plots (in the legend), where methods are sorted by this metric. Also, full numeric values for all models, scoring functions and loss functions are provided in Table 13 in Appendix B.
>
> Figure 1 presents our proposed ranking-based likelihood fine-tuning methods against a subset of baselines. This result is important and demonstrates that one can **improve** the multi-round performance of a “weaker” PLM to perform in-line with, or exceed, the recent SOTA baseline method ProteinNPT (Notin, P. et al. NeurIPS 2024).
>
>  To directly compare the fine-tuning strategies for a given PLM, regression (MSE) based PLM baselines are presented in Figure 2 (ESM-2) and Figure 3 (PoET) in Appendix B. These results demonstrate that for a given PLM, our proposed likelihood fine-tuning method is almost always preferable.
>
> Please find additional low-N experiments in Reviewer vxYr (Experimental Analysis).
>
> Baseline modeling
>
> We respectfully disagree with the reviewer on this point. ProteinNPT is a SOTA general purpose PLM developed specifically for low-data regimes. An extract from their work: “we introduce ProteinNPT, a non-parametric transformer variant tailored to protein sequences and **particularly suited to label-scarce and multi-task learning settings**”.
> We demonstrate comparable (and in some cases better) performance than ProteinNPT with “weaker” PLMs via pair-wise ranking loss functions directly applied to the likelihoods.
> Furthermore, we tuned the learning rate and the number of training steps for all models evaluated.
>
> Relation to Metalic:
>
> This is concurrent work and takes a different approach to improve **zero-shot** predictions of a PLM. Metalic is an in-context meta-learning approach that “learns to learn” to adapt to available in-context sequences available at test time. It does so via a meta-training phase over a distribution of similar fitness prediction tasks. They leverage the same subset of ProteinGym landscapes (as per ProteinNPT and our work), and specifically target zero-shot adaption.
>
> Section 6.2 is included in Section 6 to establish its contribution towards multi-round design. That is, we introduce PLM ensemble models that 1) explicitly model predictive uncertainty and 2) apply principled acquisition functions. Both advancements are aimed at performing BO in Result 2, rather than fitness prediction in Section 5.
>
> Questions for Authors:
>
> The assumption that our proposed methods achieved sub-optimal results in larger N data regimes is incorrect. Please allow us to clarify: Our motivation is driven by the multi-round design setting, where we follow the SOTA ProteinNPT experimental setup (Notin, P. et. al. NeurIPS 2023b). This consists of fine-tuning the PLM at each round and acquiring 100 new sequences per round, for a total of 10 rounds. This is a realistic biological design setting since many wet lab experimental platforms compute ~100 scores in a single “plate” in parallel. Therefore, our thesis is that improved fitness prediction at low-N datasets (as per Table 1), drives meaningful data acquisition and multi-round design performance (Table 13 and Figures 1, 2 and 3).
>
> ‘Evo tuning’
>
> This is an orthogonal research direction. Indeed, test-time training methods have been demonstrated to improve model performance using an unsupervised objective on test sequences, or indeed on homologs [1]. Also, whilst PoET conditions on MSA sequences, and demonstrates a clear benefit doing so, our work focuses on demonstrating that directly adapting model likelihoods using small experimental datasets improves fine-tuning. In practice, both or a hybrid approach could be applied.
>
> [1] Bushuiev, A. et. al. Training on test proteins improves fitness, structure, and function prediction 2025
>
> Multi-round eval:
>
> Yes, your intuition is correct. A random seed of initial data is sampled from the landscape (where the seed is fixed for all methods), so the initial training set is identical. Furthermore, we evaluate all methods across three seeds.
>
> wt-marginal strategy is a standard approach to compute the likelihood of MLM mutations introduced in Meier J., et. al (NeurIPS 2021). Popularised due to its efficiency and the widespread adoption of ESM. Concretely, the method computes the log-odds ratio of the probability of the mutated amino-acid with the wild-type's amino-acid. It does this in a single forward pass of the wt sequence.
>
> Applying the MSE loss directly to the likelihood scoring function results in unstable performance (Table 6 and Table 13 Appendix). We did not explore scaling the likelihoods.

---

> > ### Comment · Reviewer_AKmL · 2025-04-03
> >
> > Thanks for the feedback and clarifications.
> >
> >
> > I have a few follow-up questions:
> > 1) Regarding Metalic, it was unclear to how 'zero-shot' is defined here. Do you mean that the weights of the model are not updated or that no task-specific labeled examples are required to form predictions? In other words, would could your modeling technique be used in their evaluation setup and vice-versa?
> >
> > 2)  The AUC score analysis in Figure 1 is fairly informal. Are these differences statistically significant? \

---

> > > ### Author Response · Authors · 2025-04-04
> > >
> > > > could your modeling technique be used in their evaluation setup and vice-versa?
> > >
> > > In fact, **Metalic does use our likelihood ranking-based fine-tuning strategy** in both their meta-training and fine-tuning phases. This clearly shows the value and applicability of this work.
> > >
> > >
> > > > zero-shot: Do you mean that the weights of the model are not updated or that no task-specific labeled examples are required to form predictions
> > >
> > > To clarify further, both are true. In Metalic ‘zero-shot’ is defined as having no task-specific labeled sequences available to fine-tuning the PLM via weight updates. This would correspond to N=0 in our work. The authors demonstrate that by introducing a meta-training phase they can fine-tune (weight updates) their PLM over many related prediction tasks to take into consideration in-context sequences, and therefore adapt (no weight updates) to sequences provided at test time. They demonstrate improved zero-shot prediction accuracy, i.e.  Spearman correlation of predictions with respect to ProteinGym fitness scores.
> > >
> > > For completeness, Metalic goes on to introduce a ‘few-shot’ setting where N=16 or N=128 sequences (with ground truth values) are available to fine-tune (weight update) the PLM. The performance of Metalic relative to baselines reduces in the higher N data setting.
> > >
> > > >  The AUC score analysis in Figure 1 is fairly informal. Are these differences statistically significant?
> > >
> > > The key take away from Figure 1, is not whether the AUC values are significantly different from ProteinNPT (which I suspect they are not), but rather that we can take a “weaker” PLM model, e.g. ESM-2 (650M) and via a relatively straightforward fine-tuning approach, it performs competitively, and often outperforms, the SOTA design method ProteinNPT.
> > >
> > >
> > > If we have sufficiently addressed your concerns, we would kindly request that you raise your score accordingly.

---

### Official Review · Reviewer_vxYr · 2025-03-18

**Overall Recommendation:** 3

**Summary:**

This paper examines likelihood-based / rank-based finetuning for pLM, particularly for the low data fitness prediction setting. The authors formalize pairwise ranking losses for masked models (e.g. ESM-series), family/MSA-based autoregressive models (e.g. PoET), and conditional models. The results show that these methods can outperform MSE-based finetuning on frozen embeddings. They also examine ensemble strategies to leverage the context-dependent nature of PLM mutation distributions. Experiments are done on ProteinGym benchmarks.

**Claims And Evidence:**

The primary claim is that likelihood-based finetuning is better than MSE-finetuning or directly using frozen embeddings, particularly in low-data settings. This is supported by the empirical tables and results, which does consistently show that their pairwise loss adaptations yield better results, sometimes to a high degree. They also show that this can be extended to multi-round BO.

I'm not super sure how Figure 1 supports the fact that using ranking-based losses rather than MSE-based losses improves multi-round prediction. It doesn't look like that the ensemble methods are performing better? Why are non-ensemble methods used as the baseline, since the paper has thus far mostly talked about the scoring function?

I'm also curious about how N was defined for the "low-N" claim, which will be expanded up below. Generally I found this claim a bit weak, and I think it can be easily strengthened by running more experiments at more numbers of N.

**Essential References Not Discussed:**

This work [1] examining Bradley-Terry losses for fitness prediction might be relevant; the authors also find that it outperforms MSE. I think Brookes et al. has better theoretically explanations, but this current submission has more breadth in the types of models it cover.

[1] Contrastive losses as generalized models of global epistasis. Brookes et al., 2024. https://proceedings.neurips.cc/paper_files/paper/2024/file/a9b938e79504889f905d549f8d53e405-Paper-Conference.pdf

**Experimental Designs Or Analyses:**

* I don't understand the few-shot experiment setup: section 5.1 states that 32,182 or 512 sequences were randomly sampled for training and evaluated on 2000 or 5000 samples, but Table 1 includes n values of 32, 28, and 512. My first thought when reading section 5.1 was that 512 doesn't seem like a low enough data regime; for ex. Biswas et al. uses 24 as the definition of low-N, which seems closer to real-world scenarios.

[1] Low-N protein engineering with data-efficient deep learning. Biswas et al., 2021. https://www.nature.com/articles/s41592-021-01100-y

**Methods And Evaluation Criteria:**

The method involves creating pairwise ranking loss adaptations for masked language models, conditional models, and family-based autoregressive models. To do so, ProteinGym mutation landscapes are used and Spearson correlation is used for comparison.

As I'll expand upon in the Questions for Authors area, I have a few questions on the max fitness metrics, number of training data points, and observed trends.

**Other Comments Or Suggestions:**

1. I think the main claim of this working well for low-N settings would be more impactful if more $n$ was tried, and with performance perhaps plotted visually (e.g. n on x-axis and performance on y axis). The few-shot claim could be really interesting if the eval was more rigorous.
2. Error bars would help with better understanding the difference.
3. Nitpick: I'm not a fan of "Likelihood-based" in the title; I feel like "Ranking-based" would be a clearer description. Likelihood-based makes it sound like we're taking likelihoods from a larger model to explicitly finetune a smaller model or something of that sort, though this is of course a personal interpretation.
4. Nitpick: "Masked modulo" is, as far is I can tell, introduced before it was defined.

**Other Strengths And Weaknesses:**

**Strengths**: the results are strong, and similar to [1], suggests that we should move towards using ranking based losses for fitness prediction.

**Weaknesses**: The choice of using a ranking based loss is not as well-motivated with theoretical motivations in the same way as [1]. Also, since we're using a ranking based loss, I'm not sure if Spearman correlation really makes sense as the metric to use (i.e. the loss never reinforced knowing the exact _number_, only the relative _ranks_), though I could be persuaded otherwise on this point. Though the tables presented seem to support the overall claim, it feels like it should be not too hard to run more comprehensive results (e.g. across more values of N, lower values of N, maybe for more tasks or report results separately for each task.

**Overall**: Since the main contribution of this work is empirical rather than theoretical, I think the work would be stronger if the empirical results were more robust and the motivations behind the experiment design better explained. E.g. in what sorts of real-world situations might we want to use one finetuning loss over the other? What should I take away from this paper for my own research?

[1] Contrastive losses as generalized models of global epistasis. Brookes et al., 2024. https://proceedings.neurips.cc/paper_files/paper/2024/file/a9b938e79504889f905d549f8d53e405-Paper-Conference.pdf

**Questions For Authors:**

1. Do the authors have any hypotheses about why the gap between preference based training and MSE-based finetuning starts to close with higher N?
2. Could authors also report results for lower values of N, preferably with error bars?
3. What happens if we use maximum fitness rather than Spearman correlation, since often times we only care about the fitness prediction accuracy at the top end rather than the bottom end? Presumably this should be in favor of ranking based methods?
4. Clarification questions on the multi-round experiments: in Figure 1, for the non-ensemble methods, how were uncertainty estimates obtained? And what happens if we use MSE-based scoring functions as a baseline?

**Relation To Broader Scientific Literature:**

Protein fitness landscape prediction has become a canonical problem in protein ML. ProteinGym also maintains a leaderboard, which makes it easier to compare results.

Nitpick: the introduction cites Gordon et al. [1] to back up the claim that pLM likelihoods implicitly capture function/structural constraints, but I think the message of that work was actually somewhat of the opposite, namely that positional likelihoods have more to do with the likelihood over the WT sequence, which in turn "stems from the preferences encoded in the training data itself" [1]. This also indicates a possible weakness of this paper (i.e. likelihoods are not always perfect indicators of fitness), but since the likelihood <-> fitness connection has been assumed by so many other papers, it shouldn't be held as a critique unique to this work.

[1] Protein Language Model Fitness Is a Matter of Preference. Gordon et al., 2024. https://www.biorxiv.org/content/10.1101/2024.10.03.616542v1.full.pdf

**Theoretical Claims:**

n/a

---

> ### Author Rebuttal · Authors · 2025-03-31
>
> Thank you very much for your detailed feedback. If we have sufficiently addressed your concerns, we kindly ask that you raise your score accordingly, or let us know if further clarification is needed.
>
> Claims and Evidence
>
> Your comment is correct, Figure 1 does not show that. In fact, Table 13 (Appendix) demonstrates that the ranking fine-tuning counterparts outperform regression methods for multi-round design tasks. We plot MSE-based PLM baselines in Figure 2 (ESM) and Figure 3 (PoET) in Appendix B.
>
> To clarify, without modification, non-ensemble PLM methods do not estimate uncertainty. Thus in multi-round design experiments greedy acquisition strategies are used (based on the predictions only). We include ensemble baselines as they specifically provide a measure of uncertainty, allowing a principled way to trade off exploration and exploitation via (non-greedy) acquisition functions.
>
> Experimental Analysis
>
> The N in Table 1 refers to the number of sequences used for fine-tuning the PLM. We include fitness prediction results N=32, 128, and 512. Rationale for N=512 is to demonstrate (as you correctly point out the trend) that for higher N, the performance gap between ranking and regression shrinks. Our hypothesis is that in higher data regimes, the parametric heads have sufficient data to fully adapt and approximate the fitness, something which is not true in the critical low-N settings, such as N=32 or 128. We demonstrate that directly adapting the likelihoods (via ranking-based losses) in these settings achieves the best performance across model classes.
>
> We agree with your suggestion of additional low-N experiments. Please find N=24 and N=48 fitness prediction results below which additionally support our hypotheses in low-N regimes, and the main claims made in the paper. We will include the Figure as suggested. Additionally, we will add individual landscape scores into the Appendix, and include errors in Table 1.
>
> Additional N=24, N=48 single-mutant fitness prediction results
>
> ESM-2 (650M) linear-head mse loss = (0.263, 0.308) vs wt-marginals ranking = (**0.449, 0.466**)
>
> PoET linear head mse = (0.429, 0.472) vs likelihood ranking = (**0.513, 0.541**)
>
> ProteinNPT (MSAT) = (0.394,  0.461) and ProteinNPT (ESM-1v) =  (0.398, 0.422)
>
> We note it is computationally expensive to fine-tune the broad suite of PLM methods in our paper and thus we are limited in the number of additional experiments. The set of eight single mutant landscapes comprise a representative set of DMS used as validation and ablation sets in ProteinNPT (Notin, P. et. al. NeurIPS 2023b), and Metalic, (Beck, J. et. al. ICLR 2025). Furthermore, we evaluate using five challenging multi-mutant landscapes that goes beyond those used in ProteinNPT.
> Is there a particular landscape the reviewer thinks would add additional support to our analysis? If so, we will endeavour to include it in our results.
> The fitness prediction and multi-round design are standard experimental setups and follow previous works, ProteinNPT, and Kermut (Groth, P.M. et al. NeurIPS 2024).
>
> References
>
> Regarding your observation that Brookes et al., 2024. is a missing reference, please kindly note that we cite and briefly discuss their work on Page 2. They do indeed introduce a theoretical perspective relative to elucidating epistasis, but their work is limited in the models they use. Whereas, the focus of our work is specifically on providing general recommendations for **adapting the likelihoods of pre-trained PLMs**, a setting which they do not address.
> We thank the reviewer for highlighting the subtlety with regards to Gordon et al., 2024, and we will update the manuscript.
>
> Spearman
>
> We believe there is a misunderstanding here: whilst ranking-based fine-tuning parameterises a Bradly-Terry model, the underlying PLM still predicts a sequence (or mutation) likelihood. It is these likelihoods that we compute the Spearman correlation with respect to the ground truth fitness scores in ProteinGym. This is standard practice in the literature, e.g. Krause, B. et. al. 2021.
>
> Real-world situations
>
> **Our results demonstrate improved fitness prediction (Table 1) and improved multi-round design (Table 13) when leveraging ranking-based fine-tuning in low data settings across a broad range of popular PLMs**. Recently, we have seen the proliferation of pre-trained PLMs across many protein engineering tasks, and our work provides actionable insights for practitioners to get better predictive performance from existing pre-trained models, with less data.
>
> Likelihood-based title
>
> In Table 6 (Appendix) we ablate methods that directly fine-tune the model likelihood, and those that apply a parametric head to the representations. The results demonstrate that ranking-loss applied to the output of the parametric (linear) head also performs poorly. Our key recommendation is that our results support directly fine-tuning model **likelihoods** using a ranking loss function.

---

### Decision · Program_Chairs · 2025-05-01

**Decision:**

Reject

**Comment:**

The paper considers finetuning protein language models (PLMs) for protein fitness prediction.  Ranking-based loss functions are proposed to fine-tune the likelihoods of widely used PLMs including masked- and autoregressive family-based PLMs. The approach is demonstrated on supervised fitness prediction and pool-based optimization tasks.

The proposed approach is sound and well-motivated. The ensembling strategies are also very relevant. The authors have addressed several concerns and some reviewers have raised their scores as a result. However, the paper remains a borderline submission.

We strongly urge the authors to revise their manuscript to incorporate the additional results (for N=24 and N=48),  and additional clarifying comments, including relationship with related work (Metallic,Kermut)  into the manuscript. In addition, it would be valuable to include the discussion on statistical significance provided by the authors in response to Reviewer AKmL.